# Treeffuser: Probabilistic Predictions via Conditional Diffusions with Gradient-Boosted Trees

**Nicolas Beltran-Velez**[*1] **Alessandro Antonio Grande**[* 2,3] **Achille Nazaret**[* 1,3]
**Alp Kucukelbir**[1,4]               **David Blei**[1,5]

[1]Department of Computer Science, Columbia University, New York, USA
[2]Halvorsen Center for Computational Oncology, Memorial Sloan Kettering
Cancer Center, New York, USA
[3]Irving Institute for Cancer Dynamics, Columbia University, New York, USA
[4]Fero Labs, New York, USA
[5]Department of Statistics, Columbia University, New York, USA

## Abstract

Probabilistic prediction aims to compute predictive distributions rather than single point predictions. These distributions enable practitioners to quantify uncertainty, compute risk, and detect outliers. However, most probabilistic methods assume parametric responses, such as Gaussian or Poisson distributions. When these assumptions fail, such models lead to bad predictions and poorly calibrated uncertainty. In this paper, we propose Treeffuser, an easy-to-use method for probabilistic prediction on tabular data. The idea is to learn a conditional diffusion model where the score function is estimated using gradient-boosted trees. The conditional diffusion model makes Treeffuser flexible and non-parametric, while the gradient-boosted trees make it robust and easy to train on CPUs. Treeffuser learns well-calibrated predictive distributions and can handle a wide range of regression tasks—including those with multivariate, multimodal, and skewed responses. We study Treeffuser on synthetic and real data and show that it outperforms existing methods, providing better calibrated probabilistic predictions. We further demonstrate its versatility with an application to inventory allocation under uncertainty using sales data from Walmart. We implement Treeffuser in https://github.com/blei-lab/treeffuser.

## 1  Introduction

In this paper, we develop a new method for probabilistic prediction from tabular data. This problem is important to many fields, such as power generation [1], finance [2], and healthcare [3]. It drives decision processes such as supply chain planning [4], risk assessment [5, 6], and policy-making [7].

Example: Manufacturing plants measure information as they operate. This information—properties of raw materials, operational flows, temperatures, and level measurements—collectively determine the output of the plant [8]. From this data, how can manufacturers adapt operations to reduce emissions while maximizing profits? The answer requires both good predictions and estimates of uncertainty, so as to trade off the risk of failure with the reward of lower emissions and higher profit. More broadly, such industrial workflow problems often rely on predictions from vast amounts of tabular data—observations of variables arranged in a table [9, 10].

---

[*]Equal contribution, authors listed in alphabetical order.

38th Conference on Neural Information Processing Systems (NeurIPS 2024).

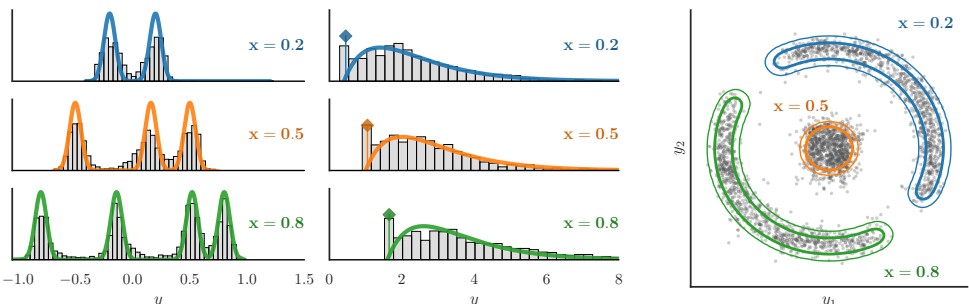

Figure 1: Samples $\boldsymbol{y} \mid \boldsymbol{x}$ from Treeffuser vs. true densities, for multiple values of $\boldsymbol{x}$ under three different scenarios. Treeffuser captures arbitrarily complex conditional distributions that vary with $\boldsymbol{x}$.

Our method builds on two ideas: diffusions [11] and gradient-boosted trees [12]. Diffusions accurately estimate conditional distributions, but existing methods are not designed for tabular data. Decision trees excel at analyzing tabular data, but do not provide probabilistic predictions. Our method, which we call *Treeffuser*, combines the advantages of both formalisms. It defines an accurate tree-based diffusion model for probabilistic prediction which can easily handle large datasets.

Fig. 1 shows Treeffuser in action. We feed it data from three complex distributions–a multimodal response with varying components, an inflated response with shifting support, and a multivariate response with dynamic correlations. For each task, Treeffuser uses trees to learn a diffusion model and then outputs samples from the target conditional. These samples fully capture the complexity of the response functions.

Treeffuser exhibits several advantages:

- It is nonparametric. It makes few assumptions about the form of the conditional response distribution. For example, it can estimate response distributions that are multivariate, multimodal, skewed, and heavy-tailed.

- It is efficient. Diffusions can be slow to train [13]. Treeffuser relies on trees and is fast. For instance, in section 5.3, Treeffuser trains from a table with 112,000 observations and 27 variables in 53 seconds on a laptop.

- It is accurate. On benchmark datasets, Treeffuser outperforms NGBoost [14], IBUG [15], quantile regression [16] and deep ensembles [17]. It provides better probabilistic predictions, including more precise quantile estimations and accurate mean predictions.

The rest of the paper is organized as follows. Section 2 reviews diffusions and gradient-boosted trees. Section 3 integrates these two ideas into Treeffuser and justifies the method. Section 4 discusses related work. Section 5 studies synthetic and standard benchmark datasets.

## 2    Background

Treeffuser combines two ideas: diffusion models and gradient-boosted trees (GBTs). This section provides a gentle introduction to both topics. Familiar readers can skip ahead to Section 3.

### 2.1    Diffusion Models

A diffusion model is a generative model that learns an arbitrarily complex distribution $\pi(\boldsymbol{y})$. It consists of two processes: forward and reverse diffusion.

The forward process takes the target distribution $\pi(\boldsymbol{y})$ and continuously transforms it into a simple distribution. It does so by progressively adding noise to samples from $\pi$:

$$\begin{cases} \text{Draw } \boldsymbol{y}(0) \sim \pi, \\ \text{Evolve } \boldsymbol{y}(t) \text{ until } T \text{ according to: } \mathrm{d}\boldsymbol{y}(t) = f(\boldsymbol{y}(t), t)\mathrm{d}t + g(t)\mathrm{d}\boldsymbol{w}(t), \end{cases} \tag{1}$$

where Eq. (1) defines a stochastic differential equation (SDE) with a standard Brownian motion $\boldsymbol{w}(t)$, and drift and diffusion functions $f$ and $g$. The time horizon $T$ is large, such that the resulting marginal distribution $p_{\text{simple}} := p_T(\boldsymbol{y}(T)) = \int_{\boldsymbol{y}'} p_T(\boldsymbol{y}(T) \mid \boldsymbol{y}(0) = \boldsymbol{y}')\pi(\boldsymbol{y}')\mathrm{d}\boldsymbol{y}'$ is agnostic to $\pi$.

The reverse process transforms the simple distribution back into the target distribution by denoising perturbed samples from $p_{\text{simple}}$. It posits the following model, which runs backward from $T$ to 0:

$$\begin{cases} \tilde{\boldsymbol{y}}(T) \sim p_{\text{simple}}, \\ \mathrm{d}\tilde{\boldsymbol{y}}_t = [f(\tilde{\boldsymbol{y}}(t), t) - g(t)^2 \nabla_{\tilde{\boldsymbol{y}}(t)} \log p_t(\tilde{\boldsymbol{y}}(t))]\mathrm{d}t + g(t)\mathrm{d}\tilde{\boldsymbol{w}}(t), \end{cases} \quad (2)$$

where $\tilde{\boldsymbol{w}}(t)$ is a standard Brownian with reversed time. Anderson [18] shows that $\tilde{\boldsymbol{y}}(t) \overset{d}{=} \boldsymbol{y}(t)$ for each time $t$, and so $\tilde{\boldsymbol{y}}(0) \sim \pi$. This means that by drawing a sample from $p_{\text{simple}}$ and then numerically approximating the reverse process in Eq. (2), we obtain samples from $\pi$.

However, the score function, $\boldsymbol{y} \mapsto \nabla_{\boldsymbol{y}} \log p_t(\boldsymbol{y})$, is usually unknown. Vincent [19] shows it can be estimated from the trajectories of the forward process as the minimizer of the following objective:

$$\left[\boldsymbol{y} \mapsto \nabla_{\boldsymbol{y}} \log p_t(\boldsymbol{y})\right] = \underset{s \in \mathcal{S}}{\arg\min}\, \mathbb{E}_t \mathbb{E}_\pi \mathbb{E}_{p_{0t}} \left[\left\|\nabla_{\boldsymbol{y}(t)} \log p_{0t}(\boldsymbol{y}(t) \mid \boldsymbol{y}(0)) - s(\boldsymbol{y}(t), t)\right\|^2\right], \quad (3)$$

where $\mathcal{S}$ is the set of all possible functions of $\boldsymbol{y}$ indexed by time $t$, and $p_{0t}$ denotes the conditional distribution of $\boldsymbol{y}(t)$ given $\boldsymbol{y}(0)$. In practice, we set $t \sim \text{Uniform}([0, 1])$ and choose the drift $f$ and diffusion function $g$ so that $p_{0t}$ is Gaussian. We detail our choice of $f$ and $g$ in Appendix C.

We approximate the expectations in Eq. (3) with the empirical distribution of the observed $\boldsymbol{y}$ for $\mathbb{E}_\pi$, and with the known Gaussian trajectories $\boldsymbol{y}(t) \mid \boldsymbol{y}(0) = \boldsymbol{y}$ for $\mathbb{E}_{p_{0t}}$. The objective can then be minimized by parametrizing the score $s$ with any function approximator, such as a neural network or, as we do, with trees. By estimating the score function, we effectively learn the distribution $\pi$.

For a more detailed introduction to diffusion models, we provide an expanded version of this section in Appendix A.

## 2.2 Gradient Boosted Trees

Consider the following common machine learning task. Given variables $(\boldsymbol{a}, b) \sim \pi$, where $b$ is scalar, learn the function $F^*(\boldsymbol{a})$ that best approximates $b$ from $\boldsymbol{a}$:

$$F^* = \underset{F}{\arg\min}\, \mathbb{E}_{(\boldsymbol{a}, b) \sim \pi}\left[L(F(\boldsymbol{a}), b)\right], \quad (4)$$

where $L$ is a loss function measuring the quality of $F$, such as the squared loss $L(u, v) = (u - v)^2$. Gradient-boosted trees (GBTs) are a widely used non-parametric machine learning model for this task [12]. GBTs use decision trees [20] as simple building block functions $f : \mathbb{R}^d \to \mathbb{R}$ to form a good approximation $\hat{F}$ of $F^*$. GBTs sequentially build approximations $\hat{F}_i$ defined as

$$\hat{F}_i(\boldsymbol{a}) = \sum_{m=0}^{i-1} \varepsilon f_m(\boldsymbol{a}), \quad (5)$$

where $\varepsilon \in (0, 1)$ is a parameter akin to a learning rate.

Each decision tree $f_i$ is constructed to minimize the squared error between the current approximation $\hat{F}_i$ and the negative gradient of the loss function:

$$f_i = \underset{f \in \text{Trees}}{\arg\min}\, \mathbb{E}_{\boldsymbol{a}, b}\left[\left(f(\boldsymbol{a}) + \frac{\partial L(F_i(\boldsymbol{a}), b)}{\partial F_i(\boldsymbol{a})}\right)^2\right]. \quad (6)$$

As the number of trees $i$ increases, the approximation $\hat{F}_i$ becomes a better minimizer of Eq. (4). Various modifications to this basic algorithm have been proposed, including different loss functions in Eq. (4), higher order optimizations for Eq. (6) and general heuristics for faster training [21–23].

## 3 Probabilistic Prediction via Treeffuser

Treeffuser tackles probabilistic prediction by learning a diffusion model with gradient-boosted trees. It is particularly well-suited to modeling tabular data. Trees offer useful inductive biases, natural

handling of categorical and missing data, and fast and robust training procedures. Diffusions eliminate the need for restrictive parametric families of distributions and protect against model misspecification.

Denote an independently distributed set of observations as $\mathcal{D} = \{(\boldsymbol{x}_i, \boldsymbol{y}_i)\}_{i=1}^n$. Treeffuser forms the predictive distribution $\pi(\boldsymbol{y} \mid \boldsymbol{x})$ as a function of inputs $\boldsymbol{x}$. We first introduce conditional diffusion models and discuss the conditional score estimation problem for both univariate and multivariate outcomes. We then outline the training and sampling procedures for Treeffuser.

## 3.1 The Conditional Diffusion Model

Treeffuser produces a distribution over $\boldsymbol{y}$ for each value of $\boldsymbol{x}$ using conditional diffusion models. Unlike approaches that guide an unconditional model to achieve conditionality, Treeffuser follows the line of work that directly fits the conditional score function [24–26].

**Conditional diffusion models.** The diffusion models introduced in Section 2.1 target the marginal distribution of $\boldsymbol{y}$. Here, we extend them to conditional distributions $\pi_{\boldsymbol{x}}(\boldsymbol{y}) := \pi(\boldsymbol{y} \mid \boldsymbol{x})$.

To model $\pi_{\boldsymbol{x}}$, assign a diffusion process $\boldsymbol{y}_{\boldsymbol{x}}(t)$ to each value of $\boldsymbol{x}$. These processes share the same diffusion equation but have different boundary conditions corresponding to their target conditionals:

$$\begin{cases} \boldsymbol{y}_{\boldsymbol{x}}(0) \sim \pi_{\boldsymbol{x}}(\boldsymbol{y}), \\ \mathrm{d}\boldsymbol{y}_{\boldsymbol{x}}(t) = f(\boldsymbol{y}_{\boldsymbol{x}}(t), t)\mathrm{d}t + g(t)\mathrm{d}\boldsymbol{w}(t). \end{cases} \tag{7}$$

As before, $f$ and $g$ are simple functions such that $\boldsymbol{y}_{\boldsymbol{x}}(T) \sim p_{\text{simple}}$ for all $\boldsymbol{x}$. Denote the marginal distribution of $\boldsymbol{y}_{\boldsymbol{x}}(t)$ as $p_{\boldsymbol{x},t}$. For two time points $t$ and $u$, where $t > u$, denote the conditional distribution $\boldsymbol{y}_{\boldsymbol{x}}(t) \mid \boldsymbol{y}_{\boldsymbol{x}}(u)$ as $p_{ut}$. (Note how $p_{ut}$ does not vary in $\boldsymbol{x}$.)

To match each $\boldsymbol{x}$-dependent forward SDE we have an $\boldsymbol{x}$-dependent reverse SDE of the form:

$$\mathrm{d}\tilde{\boldsymbol{y}}_{\boldsymbol{x}}(t) = \left[ f(\tilde{\boldsymbol{y}}_{\boldsymbol{x}}(t), t) - g(t)^2 \nabla_{\tilde{\boldsymbol{y}}_{\boldsymbol{x}}(t)} \log p_{\boldsymbol{x},t}(\tilde{\boldsymbol{y}}_{\boldsymbol{x}}(t)) \right] \mathrm{d}t + g(t)\mathrm{d}\tilde{\boldsymbol{w}}(t), \tag{8}$$

where the score function $\nabla \log p_{\boldsymbol{x},t}$ now also depends on $\boldsymbol{x}$. Similar to unconditional diffusions, by estimating the *conditional* score, we can sample from $\boldsymbol{y}_{\boldsymbol{x}}$ by first sampling $\tilde{\boldsymbol{y}}_{\boldsymbol{x}}(T)$ from $p_{\text{simple}}(\boldsymbol{y})$ and then solving the corresponding reverse SDE.

**Conditional score estimation objective.** Estimating the conditional score follows a similar strategy as the unconditional version, but with the added requirement of simultaneously estimating the score for all $\boldsymbol{x}$. Recall that by Eq. (3), for a fixed $\boldsymbol{x}$, the function $s_{\boldsymbol{x}}^*$ defined by

$$s_{\boldsymbol{x}}^* = \arg\min_{s \in \mathcal{S}} \mathbb{E}_t \mathbb{E}_{\pi_{\boldsymbol{x}}(\boldsymbol{y}_{\boldsymbol{x}}(0))} \mathbb{E}_{p_{0t}(\boldsymbol{y}_{\boldsymbol{x}}(t) \mid \boldsymbol{y}_{\boldsymbol{x}}(0))} \left[ \left\| \nabla_{\boldsymbol{y}_{\boldsymbol{x}}(t)} \log p_{0t}(\boldsymbol{y}_{\boldsymbol{x}}(t) \mid \boldsymbol{y}_{\boldsymbol{x}}(0)) - s(\boldsymbol{y}_{\boldsymbol{x}}(t), t) \right\|^2 \right]$$

satisfies $s_{\boldsymbol{x}}^*(\boldsymbol{y}_{\boldsymbol{x}}, t) = \nabla_{\boldsymbol{y}_{\boldsymbol{x}}(t)} \log p_{\boldsymbol{x},t}(\boldsymbol{y}_{\boldsymbol{x}}(t))$ for all $\boldsymbol{y}, t$ and the fixed $\boldsymbol{x}$. Intuitively, if we allow $s \in \mathcal{S}$ to also take $\boldsymbol{x}$ as input, we can gather all of these individual optimization problems into a single large problem by taking an additional expectation over $\boldsymbol{x} \sim \pi$, that is:

$$\arg\min_{S \in \mathcal{S}^+} \mathbb{E}_{\pi(\boldsymbol{x})} \mathbb{E}_t \mathbb{E}_{\pi(\boldsymbol{y}_{\boldsymbol{x}}(0))} \mathbb{E}_{p_{0t}} \left[ \left\| \nabla_{\boldsymbol{y}_{\boldsymbol{x}}(t)} \log p_{0t}(\boldsymbol{y}_{\boldsymbol{x}}(t) \mid \boldsymbol{y}_{\boldsymbol{x}}(0)) - S(\boldsymbol{y}_{\boldsymbol{x}}(t), t, \boldsymbol{x}) \right\|^2 \right]. \tag{9}$$

Here, $\mathcal{S}^+$ represents the set of functions that take $\boldsymbol{x}$ as an extra input along $\boldsymbol{y}$ and $t$. The uppercase $S$ emphasizes that $S$ takes $\boldsymbol{x}$ as input, contrary to lowercase $s$. The validity of this objective is given by the following result.

**Theorem 1** (Optimal Conditional Objective). *Define $S^*$ as the solution of Eq. (9). Then, for almost all $\boldsymbol{x}, \boldsymbol{y}, t$ with respect to $\pi(\boldsymbol{x}, \boldsymbol{y})$ and the Lebesgue measure on $t \in [0, T]$, we have*

$$S^*(\boldsymbol{y}, t, \boldsymbol{x}) = \nabla_{\boldsymbol{y}} \log p_{\boldsymbol{x},t}(\boldsymbol{y}). \tag{10}$$

We refer to Eq. (9) as *the conditional score objective*. The proof is provided in Appendix B.

## 3.2 The Trees

Treeffuser uses gradient-boosted trees (GBTs) to minimize Eq. (9). As GBTs work on scalar outputs, we separate the conditional score objective into an equivalent set of $d_y$ independent scalar-valued

| **Algorithm 1:** Treeffuser Training | **Algorithm 2:** Treeffuser Sampling |
|---|---|
| **Data:** $\mathcal{D}$, $R$, SDE-specific $(h, T)$ | **Data:** GBTs $(U_1, ..., U_{d_y})$, input instance $\boldsymbol{x}$, |
| **Result:** GBTs $(u_1, ..., u_d)$ optimizing Eq. (12) | $\quad$ discretization steps $n_d$, SDE-specific |
| $\mathcal{D}' \leftarrow \varnothing$ | $\quad (p_{\text{simple}}, T, f, g, \sigma)$ |
| **for** $(\boldsymbol{x}^{(i)}, \boldsymbol{y}^{(i)}) \in \mathcal{D}$ **do** | **Result:** A sample $\boldsymbol{y} \sim \pi(\boldsymbol{y} \mid \boldsymbol{x})$ |
| $\quad$ **for** $r = 1, ..., R$ **do** | $\delta \leftarrow T/n_d$ |
| $\quad\quad t \sim \text{Uniform}[0, T]$ | $\boldsymbol{y} \sim p_{\text{simple}}(\boldsymbol{y})$ |
| $\quad\quad \zeta \sim \mathcal{N}(0, I_{d_y})$ | $t \leftarrow T$ |
| $\quad\quad \mathcal{D}' \leftarrow \mathcal{D}' \cup ((h(\zeta, t, \boldsymbol{y}^{(i)}), t, \boldsymbol{x}^{(i)}), -\zeta)$ | **for** $i = 1, \ldots, n_d$ **do** |
| | $\quad \boldsymbol{w} \sim \mathcal{N}(0, I_{d_y})$ |
| **for** $k = 1, ..., d_y$ **do** | $\quad \tilde{f} \leftarrow f(\boldsymbol{y}, t) - g(t)^2 U(\boldsymbol{y}, t, \boldsymbol{x})/\sigma(t; \boldsymbol{y})$ |
| $\quad \mathcal{D}^k \leftarrow \{(\boldsymbol{a}, \boldsymbol{b}_k) \mid (\boldsymbol{a}, \boldsymbol{b}) \in \mathcal{D}'\}$ | $\quad \boldsymbol{y} \leftarrow \boldsymbol{y} - (\tilde{f}\delta + g(t)\delta \boldsymbol{w})$ |
| $\quad U_k \leftarrow \text{TrainGBT}(\mathcal{D}^k)$ | $\quad t \leftarrow t - \Delta$ |
| **return** $(U_1, \ldots, U_d)$ | **return** $\boldsymbol{y}$ |

sub-problems

$$S_k^* = \underset{S_k \in \text{GBT}}{\arg\min} \, \mathbb{E}_t \mathbb{E}_{\pi(\boldsymbol{x}, \boldsymbol{y})} \mathbb{E}_{p_{0t}(\boldsymbol{y}_{\boldsymbol{x}}(t) \mid \boldsymbol{y})} \left[ \left( \frac{\partial \log p_{0t}(\boldsymbol{y}_{\boldsymbol{x}}(t) \mid \boldsymbol{y})}{\partial y_{\boldsymbol{x}}(t)_k} - S_k(\boldsymbol{y}_{\boldsymbol{x}}(t), t, \boldsymbol{x}) \right)^2 \right], \qquad (11)$$

where $\boldsymbol{a}_k$ denotes the $k$-th component of vector $\boldsymbol{a}$.

Recall that the drift and diffusion functions of the forward process are chosen such that $p_{0t}$ is Gaussian. Let $m = m(t; \boldsymbol{y})$ and $\sigma = \sigma(t)$ denote the corresponding mean and standard deviation, respectively. Treeffuser replaces the partial derivative in Eq. (11) with its closed-form expression as a function of $m$ and $\sigma$. Treeffuser further reparametrizes $S(\boldsymbol{y}, t, \boldsymbol{x})$ with $U(\boldsymbol{y}, t, \boldsymbol{x})/\sigma(t)$ and defines $h(\zeta, t, \boldsymbol{y}) = m(t; \boldsymbol{y}) + \zeta\sigma(t)$, the process by which a sample $\boldsymbol{y}$ at time 0 gets diffused into a sample at time $t$ with Gaussian noise $\zeta$. The optimization problems in Eq. (11) are then:

$$\forall k \in \{1, ..., d_y\}, \quad U_k^* = \underset{U_k}{\arg\min} \, \mathbb{E}_{(\boldsymbol{x}, \boldsymbol{y}) \sim \pi} \mathbb{E}_t \mathbb{E}_{\zeta \sim \mathcal{N}(0, I_{d_y})} \left[ (\zeta_k + U_k(h(\zeta, t, \boldsymbol{y}), t, \boldsymbol{x}))^2 \right]. \quad (12)$$

The next theorem justifies how the individual problems in Eq. (12) estimate the conditional score.

**Theorem 2** (Treeffuser One-Dimensional Objectives). *Denote* $U^* = (U_1^*, ..., U_{d_y}^*)$. *Then for almost all* $\boldsymbol{x}, \boldsymbol{y}, t$ *with respect to* $\pi(\boldsymbol{x}, \boldsymbol{y})$ *and the Lebesgue measure on* $t \in [0, T]$, *we have*

$$\nabla_{\boldsymbol{y}} \log p_{\boldsymbol{x}, t}(\boldsymbol{y}) = \frac{U^*(\boldsymbol{y}, t, \boldsymbol{x})}{\sigma(t)}. \quad (13)$$

Each problem in Eq. (12) is a GBT problem where the notation within Eq. (4) corresponds to $F := U_k$, $\boldsymbol{a} := (h(\zeta, t, \boldsymbol{y}), t, \boldsymbol{x}) \in \mathbb{R}^{d_y + 1 + d_x}$, $b := -\zeta_k \in \mathbb{R}$ and $L$ is the square loss. We note that the noise scaling reparametrization, $S = U/\sigma$, is key to stabilizing the learning process; see the ablation study in Appendix G.

Finally, Treeffuser approximates the expectations in Eq. (12) with Monte Carlo sampling. For each sample $(\boldsymbol{x}, \boldsymbol{y})$ from the dataset $\mathcal{D}$, Treeffuser samples $R$ pairs of $(t, \zeta) \sim \text{Uniform}([0, 1]) \otimes \mathcal{N}(0, I_{d_y})$ and creates new datasets $\mathcal{D}^k$ containing $R \cdot n$ datapoints of the form $\left( (h(\zeta, t, \boldsymbol{y}), t, \boldsymbol{x}), -\zeta_k \right)$, one $\mathcal{D}^k$ per dimension of $\boldsymbol{y}$. Then, each of these datasets is given to a standard GBT algorithm, such as LightGBM [22] or XGBoost [21]. Our implementation of Treeffuser uses LightGBM. Algorithm 1 details this procedure.

### 3.3 Sampling and Probabilistic Predictions

Treeffuser provides samples from the probabilistic predictive distribution $\pi(\boldsymbol{y} \mid \boldsymbol{x})$. It does so as in standard unconditional models by plugging in the GBT-estimated conditional score from Eq. (12)

into a numerical approximation of the SDE in Eq. (8). While Treeffuser is compatible with any SDE solver, our implementation leverages Euler-Maruyama [27] due to its good balance between accuracy and the number of function evaluations. In general, we found good performance with as few as 50 steps. Algorithm 2 implements this procedure.

The samples generated by Treeffuser can then be used to estimate means, quantiles, probability intervals, expectations, or any other quantity of interest that depends on the response distribution.

### 3.4 Limitations

The design of Treeffuser offers advantages in terms of usability, versatility, and robustness. But it also comes with a few limitations. First, the diffusion process is theoretically defined to only model continuous responses $y$. However, count and other forms of discrete responses are common in the probabilistic modeling of tabular data. While our experiments show that this limitation does not prevent Treeffuser from outperforming comparable methods on these kinds of data, there may be opportunities for further improvement with direct modeling of discrete outcomes. Second, Treeffuser does not offer a closed-form density and must solve an SDE to generate samples. This sampling process, in contrast with the fast training, can become expensive when many samples per datapoint $x$ are required.

## 4 Related Work

Treeffuser builds on advances in diffusion models to form probabilistic predictions from tabular data.

**Diffusion models.** Diffusion models excel at learning complex unconditional distributions on a range of data, such as images [28–30], molecules [31], time series [32], and graphs [33]. A common task is conditional generation, where the goal is to generate samples from a distribution conditioned on features. There are two approaches to this objective. One approach is to use guidance methods by which the score of an unconditional diffusion model is altered during generation to mimic the score of the conditional distribution [11, 34, 35]. This approach is especially popular for inverse problems [11, 36, 37]. Another approach is to train a conditional model from the start, incorporating the conditioning information during training [24–26, 38, 39]. This is the approach adopted by Treeffuser.

Treeffuser contributes to a recent line of work that applies diffusions to tabular data. This includes deep learning approaches for data generation [40], probabilistic regression [41], and missing data imputation [42]. Among these methods, CARD [41] uses neural-net based diffusions for probabilistic predictions and thus is most similar to Treeffuser in scope. We attempted to include it in our experiments using the implementation from Lehmann [43], but returned very poor results. We therefore excluded it from our testbed. The imputation of missing data has been recently extended to gradient-boosted trees [44].

**Probabilistic prediction for tabular data** Probabilistic prediction for tabular data can be classified into parametric and non-parametric methods based on their assumptions about the likelihood shape. Parametric tree-based methods include NGBoost [14] and PBGM [45]. NGBoost uses natural gradients to optimize a scoring rule, while PBGM sequentially updates the mean and standard deviation for predictions. DRFs obtain maximum likelihood estimates by using this criteria to choose splits. Neural-based parametric methods include Bayesian Neural Networks [46], MC Dropout [47], and Deep Ensembles [17]. Notably these methods are all indirectly or directly Bayesian. Another approach, normalizing flows, transforms a latent distribution via an invertible neural network [48] and has been applied to tabular data [49]. Non-parametric methods are often tree-based, such as Quantile Regression Forests [16], Distributional Random Forests (DRF) [50], and IBUG [15]. Quantile Regression Forests approximate the inverse cumulative distribution function by minimizing pinball loss, while DRF and IBUG use a tree-based similarity metric to weight training data for predictions. These methods are baselines in our empirical studies, with detailed descriptions in Appendix F.1.

## 5 Empirical studies

We demonstrate Treeffuser across three settings: synthetic data, standard UCI datasets [51], and sales data from Walmart. We find that Treeffuser outperforms state-of-the-art methods [15, 17, 50];

| Dataset | $N, d_x, d_y$ | Deep ensembles | NGBoost (Gaussian) | iBUG (XGBoost) | Quantile regression | DRF | Treeffuser | Treeffuser (no tuning) | |
|---|---|---|---|---|---|---|---|---|---|
| bike | 17,379,12,1 | **1.61 ± 0.05** | 7.15 ± 0.18 | 1.88 ± 0.11 | 1.85 ± 0.07 | 2.15 ± 0.06 | **1.60 ± 0.05** | 1.64 ± 0.05 | $\times 10^1$ |
| energy | 768, 8, 2 | 5.00 ± 0.71 | 4.78 ± 0.49 | NA | NA | 5.43 ± 0.69 | **3.07 ± 0.40** | **3.32 ± 0.48** | $\times 10^0$ |
| kin8nm | 8,192, 8, 1 | **3.59 ± 0.12** | 9.48 ± 0.29 | 7.74 ± 0.41 | 6.63 ± 0.16 | 9.44 ± 0.20 | 5.89 ± 0.14 | **5.88 ± 0.17** | $\times 10^{-2}$ |
| movies | 7,415, 9, 1 | 2.94 ± 0.35 | × | 3.42 ± 0.52 | 7.90 ± 0.68 | 5.57 ± 0.61 | **2.68 ± 0.31** | **2.69 ± 0.28** | $\times 10^7$ |
| naval | 11,934,17,1 | 4.11 ± 0.39 | 4.43 ± 0.19 | 3.20 ± 0.17 | 16.86 ± 2.46 | 4.55 ± 0.16 | **2.02 ± 0.08** | **2.46 ± 0.07** | $\times 10^{-4}$ |
| news | 39,644,58,1 | 2.53 ± 0.27 | × | 3.89 ± 1.19 | 2.32 ± 0.17 | **1.98 ± 0.16** | 1.98 ± 0.17 | **1.98 ± 0.16** | $\times 10^3$ |
| power | 9,568, 4, 1 | 2.06 ± 0.10 | 2.01 ± 0.13 | 1.61 ± 0.07 | 5.40 ± 0.12 | 1.90 ± 0.11 | **1.49 ± 0.07** | **1.52 ± 0.07** | $\times 10^0$ |
| superc. | 21,263,81,1 | 4.89 ± 0.31 | 5.24 ± 0.50 | 4.14 ± 0.28 | 3.79 ± 0.14 | 4.32 ± 0.16 | **3.52 ± 0.13** | **3.60 ± 0.15** | $\times 10^0$ |
| wine | 6,497, 12, 1 | 3.59 ± 0.10 | 3.82 ± 0.11 | 3.25 ± 0.14 | 3.24 ± 0.14 | 3.30 ± 0.12 | **2.59 ± 0.13** | **2.67 ± 0.13** | $\times 10^{-1}$ |
| yacht | 308, 6, 1 | 4.86 ± 1.38 | 3.67 ± 1.47 | 3.75 ± 1.24 | 3.53 ± 1.30 | 7.73 ± 2.43 | **3.11 ± 0.99** | **3.39 ± 0.97** | $\times 10^{-1}$ |

Table 1: CRPS (lower is better) by dataset and method. × indicates the method failed to run, and NA that the method is not directly applicable to multivariate outputs. Standard deviations are measured with 10-fold cross-validation. For each dataset, the two best methods are bolded. Treeffuser provides the most accurate probabilistic predictions, even with default hyperparameters (no tuning). Powers of ten are factorized out of each row in the rightmost column.

it can capture complex distributions with multimodal, inflated, or multivariate responses; it achieves better probabilistic predictions on many real-world datasets and it produces more accurate sales forecasts on Walmart data. We also find that Treeffuser can outperform other (tuned) methods without dataset-specific hyperparameter tuning. We provide the code for Treeffuser at https://github.com/blei-lab/treeffuser.

## 5.1 Probabilistic prediction on synthetic data

We show the flexibility of Treeffuser on three probabilistic regression tasks that are difficult to model with standard parametric models. We also show its competitive performance on Gaussian data when compared to methods that posit a Gaussian likelihood.

**Treeffuser vs. complex response functions.** We design three difficult probabilistic regression tasks: a multimodal response where the number of modes changes with $x$, an inflated response with support that changes with $x$, and a multivariate response with nontrivial covariance structure. The data generation processes for these datasets are detailed in Appendix E. For each dataset, we generate 10,000 observations $(x_i, y_i)$ and train Treeffuser with its default parameters (see Appendix C).

Fig. 1 shows the histograms of 1,000 Treeffuser samples for three different values of $x$ and compares them with the true conditional densities of the response. Treeffuser captures the conditional distribution for all values of $x$ across all settings. For multimodal data, Treeffuser correctly detects the modes of the distribution without knowing their number a priori. For inflated data, it recovers the peak at the inflation point and the $x$-dependent support of the response. For multivariate data, it recovers the complex covariance structures.

**Treeffuser vs. parametric oracles.** We simulate Gaussian responses with a linear response function and compare Treeffuser to other parametric methods that assume a Gaussian likelihood. These methods serve as oracles in these experiments as they are informed by the true functional form of the conditional response distribution. Results and further details are reported in Appendix E. Among all methods, Treeffuser consistently ranks second best, only behind the Deep Ensemble oracle.

## 5.2 Probabilistic prediction on real-world datasets

We compare Treeffuser with state-of-the-art methods for probabilistic predictions on standard UCI datasets [51]. A detailed description of the baseline models can be found in Appendix F.1.

**Metrics.** We evaluate probabilistic predictions with the continuous ranked probability score (CRPS) and standard accuracy metrics.

CRPS is defined as $\mathrm{CRPS}(F, y) = \int_{-\infty}^{\infty} (F(y') - \mathbb{1}(y \leq y'))^2 dy'$, where $F$ is the cumulative distribution function of the predicted distribution $p(y \mid x)$ and $y$ is the true observed value [52]. CRPS is a proper scoring rule, in that its expected value is minimized under the true conditional distribution of the response [53]. CRPS is usually preferred over the log-likelihood, which is also

a proper scoring rule, but is sensitive to the estimation of the tail densities [54]. Also, CRPS can readily be estimated from samples of $p(y \mid \boldsymbol{x})$, which is our setting with the non-parametric methods we evaluate. We evaluate CRPS by generating 100 samples from $p(y \mid \boldsymbol{x})$ for each $\boldsymbol{x}$. For evaluating multivariate responses, we report the average marginal CRPS over each dimension.

We also measure the quality of point predictions for each model. This is the ability to predict conditional means $\mathbb{E}[y \mid \boldsymbol{x}]$. We approximate $\mathbb{E}[y \mid \boldsymbol{x}]$ using 50 samples and evaluate the accuracy using the mean absolute error (MAE) and the root mean squared error (RMSE).

**Experimental setup.**  We performed 10-folds cross-validation. For each fold, we tuned the hyper-parameters of the methods using Bayesian optimization for 25 iterations, using 20% of the current fold's training data as a validation set. Additional Bayesian optimization's iterations did not change the results. We detail the search space of hyperparameters for each method in Appendix F.2.

**Results.**  Table 1 presents CRPS by dataset and method. Treeffuser consistently provides the most accurate predictions across datasets, as measured by CRPS. Notably, it outperforms other methods even when initialized with its default parameters. There is no consistent runner-up: among parametric methods, deep ensemble does overall better than NGBoost and IBUG; quantile regression does well on some datasets but underperforms in others.

We find that Treeffuser also returns the best point predictions of $\mathbb{E}[y \mid \boldsymbol{x}]$, as reported in Table 4. For comparison, we report the accuracy of point predictions from vanilla XGBoost and LightGBM in Table 6 in Appendix F.3. These methods do not provide probabilistic predictions but are tailored for point predictions. As expected, XGBoost and LightGBM outperform or tie with all the probabilistic methods. In particular, they often tie with Treeffuser, suggesting that Treeffuser provides probabilistic prediction without sacrificing average point predictions.

Finally, we conducted an ablation study to investigate the impact of the noise-scaling reparametrization of the score function on Treeffuser's performance. As detailed in Appendix G, noise scaling is key to achieving top accuracy and stability.

### 5.3 Sales forecasting with long tails and count data

| model
metric | Deep
ensembles | IBUG | NGBoost Poisson | Quantile
regression | Treeffuser | Treeffuser
(no tuning) | |
|---|---|---|---|---|---|---|---|
| CRPS | 7.05 | 7.75 | 6.86 | 7.11 | **6.44** | **6.62** | $\times 10^{-1}$ |
| RMSE | **2.03** | 2.16 | 2.33 | 2.88 | **2.09** | **2.09** | $\times 10^{0}$ |
| MAE | **0.97** | 1.04 | **0.99** | 1.01 | **0.99** | **0.99** | $\times 10^{0}$ |

Table 2: Walmart dataset metrics (lower is better). The evaluation is on the last 30 days of data. Treeffuser provides the best probabilistic predictions alongside NGBoost Poisson. Deep ensembles excel at point predictions. Powers of tens are factorized out of each row in rightmost column.

We further demonstrate the applicability of Treeffuser on a publicly available dataset [55] for sales forecasting under uncertainty. The goal is to forecast the number of units sold for a product given features such as its price, its type (e.g., food, cloths), and past sales.

This task is challenging due to zero inflation and long tails in the distribution of item sales. (E.g., umbrella sales are typically low but can spike during rainy weeks.) It is even more challenging for a diffusion model like Treeffuser, which is designed for continuous responses and not count data.

We use five years of sales data from ten Walmart stores (a large American retail chain). We randomly select 1,000 products, training on 112,000 data points from the first 1,862 days and evaluating 10,000 other data points for the remaining 30 days.

In addition to the previous baselines, we include NGBoost Poisson, a parametric model specifically designed for count data. We evaluate the predictions returned by each method in three ways.

**CRPS and accuracy metrics.**  First, we compute the same evaluation metrics as in the previous experiments. We benchmark against the methods in the experiment and the methods in section 5.2.

The results are reported in Table 2. We find that Treeffuser again proves competitive, achieving a better CRPS than all methods and comparable MAE and RMSE.

**Posterior predictive checks.**    Second, we perform held-out predictive checks [56], examining six statistics on the number of items sold: the total count of zero sales, the highest sales figure, and sales figures at the 0.99, 0.999, and 0.9999 quantiles (lower quantiles are well captured by all methods). We produce probabilistic predictions of these quantities by returning their empirical distributions as induced by the samples generated by the models. Fig. 2 compares the observed values against the probabilistic predictions. Treeffuser best captures the proportion of zeros and performs as well as NGBoost-Poisson in modeling the behavior of the tails.

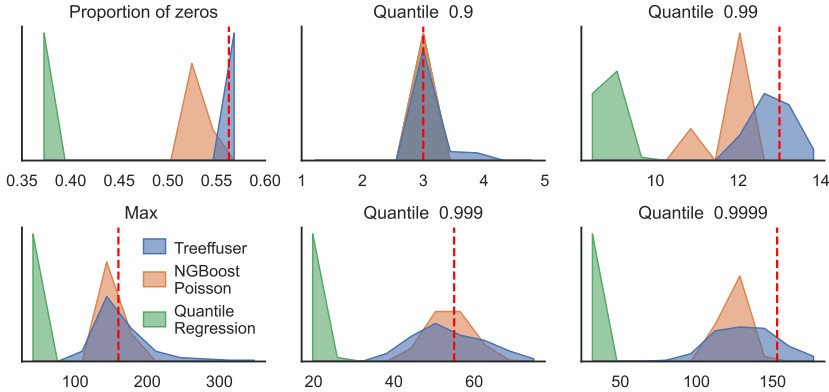

Figure 2: Posterior predictive checks for Treeffuser, NGBoost Poisson, and quantile regression. Red dashed line shows the realized value on the test set. Treeffuser best captures the inflation point at zero and performs well on the tails.

**Newsvendor model.**    Finally, we illustrate the practical relevance of accurate probabilistic predictions with an application to inventory management, using the newsvendor model [57]. Assume that every day we decide how many units $q$ of an item to buy. We buy at a cost $c$ and sell at a price $p$. However, the demand $y$ is random, introducing uncertainty in our decision. The goal is to maximize the expected profit:

$$\max_q p \, \mathbb{E}\left[\min(q, y)\right] - cq.$$

The solution to the newsvendor problem is to buy $q = F^{-1}\left(\frac{p-c}{p}\right)$ units, where $F^{-1}$ is the quantile function of the distribution of $y$.

We apply this model to evaluate the inventory decisions induced by each method on the Walmart dataset. To compute profits, we use the observed prices and assume a margin of 50% over all products. We let Treeffuser, NGBoost-Poisson, and quantile regression learn the conditional distribution of the demand of each item, estimate their quantiles, and thus determine the optimal quantity to buy.

Fig. 3 plots the cumulative profits over the last 30 days of data. Treeffuser outperforms quantile regression by a large margin and performs comparably to NGBoost-Poisson. This is coherent with our PPC results in Fig. 2, showing better quantile estimations for Treeffuser. This demonstrates that Treeffuser delivers competitive probabilistic predictions even for count data responses, a scenario it was not specifically designed to handle.

## 5.4    Runtime Performance Overview

We measure Treeffuser's performance in terms of training and inference speed across different datasets. On the M5 dataset, using default parameters, Treeffuser completed training in 53.2 seconds on a MacBook Pro M3 Max and generated 10,000 samples (one per test data point) in 2.53 seconds. We conducted additional benchmarking experiments on both the UCI and M5 datasets, with results detailed in Appendix H. (Further discussion on the model's time complexity is also provided in that appendix.) Details about the computational resources used in all of our experiments are available in Appendix D.

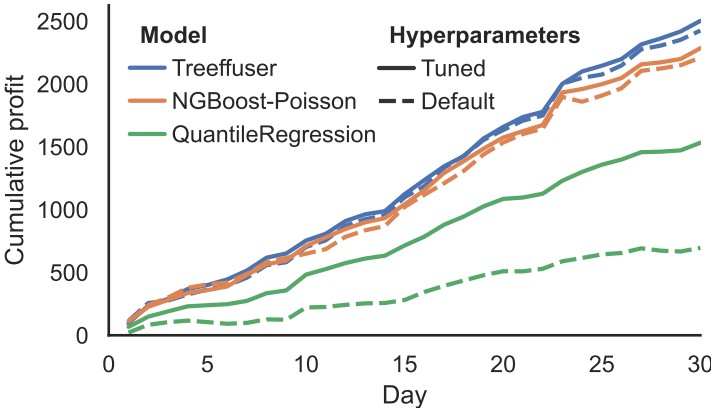

Figure 3: Cumulative profits by method on an inventory management problem. Treeffuser produces more accurate probabilistic predictions yielding higher profits.

## 6 Conclusion

We have introduced Treeffuser, a new model for probabilistic prediction from tabular data.

Treeffuser combines conditional diffusions models with gradient-boosted trees. It can capture arbitrarily complex distributions without requiring any data-specific modeling or tuning. It is amenable to fast CPU learning and naturally handles categorical data and missing values. We have demonstrated that Treeffuser outperforms state-of-the-art methods in probabilistic regression across datasets and metrics. These characteristics make Treeffuser a flexible, easy-to-use, and robust model for probabilistic predictions.

One limitation of our diffusion-based approach is the need to numerically solve an SDE to generate samples, which can be costly when producing many samples. Recent advances, such as progressive distillation [58] and consistency models [59], address this issue. Applying these methods to Treeffuser is a direction for future work.

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

## Acronyms

**CRPS**  continuous ranked probability score. 7, 8, 9

**GBT**  gradient-boosted tree. 2, 3, 4, 5, 16

**MAE**  mean absolute error. 8, 9

**RMSE**  root mean squared error. 8, 9

**SDE**  stochastic differential equation. 3, 4, 6, 14, 15, 16

# A   A primer on unconditional diffusion models

A diffusion model is a generative model that consists of two processes, a forward diffusion, and a reverse diffusion. The forward process takes an unknown target distribution $\pi(\boldsymbol{y})$ and continuously transforms it into a simple distribution $p_{\text{simple}}(\boldsymbol{y})$, typically a Gaussian. It does so by progressively adding noise to samples from $\pi$, such as our data.

The reverse process transforms the simple distribution back into the target distribution by denoising any perturbed samples from $p_{\text{simple}}$. If we can learn the reverse process, we can effectively access the target distribution $\pi$: first draw samples from $p_{\text{simple}}$ and then use the reverse transformation, as guided by the score, to map them back to $\pi$.

One way to learn this noising-denoising process is to estimate the *score function*, the gradient of the log probability of the noisy data with respect to the data itself.

We detail the three main components of score-based diffusions–the forward process, the reverse process, and the estimation of the score function–below.

**The forward diffusion process.**   A diffusion model is described by the following generative process:

$$\begin{cases} \text{Draw } \boldsymbol{y}(0) \sim \pi, \\ \text{Evolve } \boldsymbol{y}(t) \text{ until } T \text{ according to: } \mathrm{d}\boldsymbol{y}(t) = f(\boldsymbol{y}(t), t)\mathrm{d}t + g(t)\mathrm{d}\boldsymbol{w}(t), \end{cases} \tag{14}$$

where the process of evolving the stochastic differential equation (SDE) involves a standard Brownian motion $\boldsymbol{w}(t)$ and model parameters $f : \mathbb{R}^{d_y} \times [0, T] \to \mathbb{R}^{d_y}$ and $g : [0, T] \to \mathbb{R}$, respectively called the *drift* and *diffusion* functions.

The SDE in Eq. (14) determines the evolution of $\boldsymbol{y}(t)$ over time, thereby inducing a distribution of $\boldsymbol{y}(t)$ at each $t$, that we write $p_t(\boldsymbol{y}(t))$. We further write the distribution of $\boldsymbol{y}(t)$ conditional on another $\boldsymbol{y}(u)$ as $p_{ut}(\boldsymbol{y}(t) \mid \boldsymbol{y}(u))$. In practice, $f$ and $g$ are simple functions such that $p_{0t}(\boldsymbol{y}(t) \mid \boldsymbol{y}(0))$ has a closed form. In fact, if the drift $f$ is affine in $\boldsymbol{y}(t)$ and the diffusion function $g$ is a scalar that does not depend on $\boldsymbol{y}(t)$, the distribution $p_{0t}$ is always Gaussian, with a mean we write $m(t; \boldsymbol{y}(0))$ and a covariance multiple of the identity we write $\sigma(t)^2 I$.

W choose the time horizon $T$ such that the resulting marginal distribution $p_T(\boldsymbol{y}(T)) = \int_{\boldsymbol{y}'} p_T(\boldsymbol{y}(T) \mid \boldsymbol{y}(0) = \boldsymbol{y}')\pi(\boldsymbol{y}')\mathrm{d}\boldsymbol{y}'$ is a simple distribution $p_{\text{simple}}$ agnostic to $\pi$. The intuition is that as the diffusion progresses, the initial density $\pi$ is forgotten.

We detail our choice of $f$, $g$ and $T$ in Appendix C, which is the variance exploding SDE (VESDE) in Song et al. [11]. This choice induces a trivial closed form expression for the functions $m$ and $\sigma$.

There are no model parameters to learn in the diffusion model, and the conditional distributions $p_{0t}(\boldsymbol{y}(t) \mid \boldsymbol{y}(0))$ are trivial to compute. The challenge is to learn how to compute (or sample from) the conditional distribution $p_{Tt}(\boldsymbol{y}(t) \mid \boldsymbol{y}(T))$. This is called *reversing* the diffusion process. Knowing $p_{Tt}$ is key to learning the target distribution $\pi$, which can be written as $\pi(\boldsymbol{y}(0)) = \mathbb{E}_{p_T}[p_{T0}(\boldsymbol{y}(0) \mid \boldsymbol{y}(T))]$ where $p_T = p_{\text{simple}}$ is known.

**The reverse diffusion process.**   Consider the following SDE that is reversed in time from $T$ to $0$,

$$\begin{cases} \tilde{\boldsymbol{y}}(T) \sim p_{\text{simple}}, \\ \mathrm{d}\tilde{\boldsymbol{y}}_t = [f(\tilde{\boldsymbol{y}}(t), t) - g(t)^2 \nabla_{\tilde{\boldsymbol{y}}(t)} \log p_t(\tilde{\boldsymbol{y}}(t))]\mathrm{d}t + g(t)\mathrm{d}\tilde{\boldsymbol{w}}(t), \end{cases} \tag{15}$$

where $\tilde{\boldsymbol{w}}(t)$ is a standard Brownian with reversed time and $p_t$ is the density of $\boldsymbol{y}(t)$ defined by the forward SDE (1). Similar to the forward case, the reverse SDE induces a distribution $\tilde{p}_t$ for $\tilde{\boldsymbol{y}}(t)$ at each $t$. Anderson [18] shows that, if we denote by $\tilde{p}_{Tt}$ the conditional distribution of $\tilde{\boldsymbol{y}}(t)$ given $\tilde{\boldsymbol{y}}(T)$, then $\tilde{p}_{Tt} = p_{Tt}$ for any time $t$. In other words, $\tilde{\boldsymbol{y}}(t) \mid \tilde{\boldsymbol{y}}(T) = \boldsymbol{a}$ and $\boldsymbol{y}(t) \mid \boldsymbol{y}(T) = \boldsymbol{a}$ have the same distribution for all $\boldsymbol{a}$. Since we designed the diffusion such that $p_T \approx p_{\text{simple}} = \tilde{p}_T$, we have in particular $\tilde{\boldsymbol{y}}(t) \sim \boldsymbol{y}(t)$.

SDE solvers let us sample from distributions that are solutions of an SDE; hence, we can obtain samples of $\pi(\boldsymbol{y})$ by solving Eq. (15). However, the function $\boldsymbol{y} \mapsto \nabla_{\boldsymbol{y}} \log p_t(\boldsymbol{y})$, called the *score*, is usually unknown and needs to be estimated.

**Estimating the score.** Vincent [19] shows that the score can be estimated from trajectories of the SDE as the minimizer of the following objective function:

$$[\boldsymbol{y} \mapsto \nabla_{\boldsymbol{y}} \log p_t(\boldsymbol{y})] = \arg\min_{s \in \mathcal{S}} \mathbb{E}_t \mathbb{E}_\pi \mathbb{E}_{p_{0t}} \left[ \left\| \nabla_{\boldsymbol{y}(t)} \log p_{0t}(\boldsymbol{y}(t) \mid \boldsymbol{y}(0)) - s(\boldsymbol{y}(t), t) \right\|^2 \right], \quad (16)$$

where $\mathcal{S} = \{s : \mathbb{R}^{d_y} \times [0, T] \to \mathbb{R}^{d_y}\}$ is the set of all possible score functions indexed by time $t$, and $\mathbb{E}_t$ can be an expectation over any distribution of $t$ whose support is exactly $[0, T]$. Since $p_{0t}$ is a normal distribution $\mathcal{N}(m(t; \boldsymbol{y}(0)), \sigma(t)^2 I)$, we have

$$\nabla_{\boldsymbol{y}(t)} \log p_{0t}(\boldsymbol{y}(t) \mid \boldsymbol{y}(0)) = \frac{m(t; \boldsymbol{y}(0)) - \boldsymbol{y}(t)}{\sigma(t)^2}. \quad (17)$$

In practice, the expectations in Eq. (16) are approximated using the empirical distribution formed by the observed $\boldsymbol{y}$ for $\mathbb{E}_\pi$, and by sampling the simple and known forward SDE trajectories $\boldsymbol{y}(t) \mid \boldsymbol{y}(0) = \boldsymbol{y}$ for $\mathbb{E}_{p_{0t}}$. The objective can then be minimized by parametrizing $s$ with any function approximator, e.g., a neural network. The score estimator can then be used to reverse the diffusion process and estimate the target distribution $\pi(\boldsymbol{y})$.

## B  Proofs of the main theorems

In this section, we provide the proofs for the two main theorems in the text.

**Theorem 1** (Optimal Conditional Objective). *Define $S^*$ as the solution of Eq. (9). Then, for almost all $\boldsymbol{x}, \boldsymbol{y}, t$ with respect to $\pi(\boldsymbol{x}, \boldsymbol{y})$ and the Lebesgue measure on $t \in [0, T]$, we have*

$$S^*(\boldsymbol{y}, t, \boldsymbol{x}) = \nabla_{\boldsymbol{y}} \log p_{\boldsymbol{x}, t}(\boldsymbol{y}). \quad (10)$$

*Proof.* For conciseness, define for any function $s : \mathbb{R}^{d_y} \times [0, T] \to \mathbb{R}^{d_y}$ the quantity: $r(\boldsymbol{x}, s) = \mathbb{E}_t \mathbb{E}_{\pi(\boldsymbol{y}_{\boldsymbol{x}}(0))} \mathbb{E}_{p_{\boldsymbol{x}, 0t}} \left[ \left\| \nabla_{\boldsymbol{y}_{\boldsymbol{x}}(t)} \log p_{0t}(\boldsymbol{y}_{\boldsymbol{x}}(t) \mid \boldsymbol{y}_{\boldsymbol{x}}(0)) - s(\boldsymbol{y}_{\boldsymbol{x}}(t), t, \boldsymbol{x}) \right\|^2 \right]$. Also write for any function $S : \mathbb{R}^{d_y} \times [0, T] \times \mathbb{R}^{d_x} \to \mathbb{R}^{d_y}$ and $\boldsymbol{x} \in \mathbb{R}^{d_x}$ the function $S_{\boldsymbol{x}} : (\boldsymbol{y}, t) \mapsto S(\boldsymbol{y}, t, \boldsymbol{x})$ .

The function $S^*$ is characterized as, $S^* \in \arg\min_{S \in \mathcal{S}^+} \mathbb{E}_{\pi(\boldsymbol{x})}[r(\boldsymbol{x}, S_{\boldsymbol{x}})]$ where $\mathcal{S}^+ = \{S : \mathbb{R}^{d_y} \times [0, T] \times \mathbb{R}^{d_x} \to \mathbb{R}^{d_y}\}$

Define $\Omega = \{\boldsymbol{x} \in \mathbb{R}^{d_x} \mid S^*(\boldsymbol{y}, t, \boldsymbol{x}) \notin \arg\min_s r(\boldsymbol{x}, s)\}$, we will show that $\pi(\Omega) = 0$.

By [19], we know that $\nabla_{\boldsymbol{y}} \log p_{\boldsymbol{x}, t}(\boldsymbol{y}) \in \arg\min_s r(\boldsymbol{x}, s)$ for all $\boldsymbol{x}$, so by definition of $\Omega$, we have:

$$\forall \boldsymbol{x} \in \Omega, \quad r(\boldsymbol{x}, S_{\boldsymbol{x}}^*) > r(\boldsymbol{x}, (\boldsymbol{y}, t) \mapsto \nabla_{\boldsymbol{y}} \log p_{\boldsymbol{x}, t}(\boldsymbol{y})). \quad (18)$$

We further have for any other $x \notin \Omega$, $r(\boldsymbol{x}, S_{\boldsymbol{x}}^*) = r(\boldsymbol{x}, (\boldsymbol{y}, t) \mapsto \nabla_{\boldsymbol{y}} \log p_{\boldsymbol{x}, t}(\boldsymbol{y}))$.

If $\pi(\Omega) > 0$, then integrating Eq. (18) over $\pi(x)$ will yield $\mathbb{E}_{\pi(\boldsymbol{x})}[r(\boldsymbol{x}, S_{\boldsymbol{x}}^*)] > \mathbb{E}_{\pi(\boldsymbol{x})}[r(\boldsymbol{x}, ((\boldsymbol{y}, t, \boldsymbol{x}) \mapsto \nabla_{\boldsymbol{y}} \log p_{\boldsymbol{x}, t}(\boldsymbol{y}))_{\boldsymbol{x}})]$, which is not possible by definition of $S^*$. Hence $\pi(\Omega) = 0$.

Hence we have $S_{\boldsymbol{x}}^* \in \arg\min_s r(\boldsymbol{x}, s)$ for almost any $x$, so by [19] again, we can conclude that $S^*(\boldsymbol{y}, t, \boldsymbol{x}) = \nabla_{\boldsymbol{y}} \log p_{\boldsymbol{x}, t}(\boldsymbol{y})$ for almost any $\boldsymbol{x}, t, \boldsymbol{y}$.

$\square$

**Theorem 2** (Treeffuser One-Dimensional Objectives). *Denote $U^* = (U_1^*, ..., U_{d_y}^*)$. Then for almost all $\boldsymbol{x}, \boldsymbol{y}, t$ with respect to $\pi(\boldsymbol{x}, \boldsymbol{y})$ and the Lebesgue measure on $t \in [0, T]$, we have*

$$\nabla_{\boldsymbol{y}} \log p_{\boldsymbol{x}, t}(\boldsymbol{y}) = \frac{U^*(\boldsymbol{y}, t, \boldsymbol{x})}{\sigma(t)}. \quad (13)$$

*Proof.* By definition in Eq. (12), we have

$$U_k^* = \arg\min_{U_k} \mathbb{E}_{\boldsymbol{x}, \boldsymbol{y} \sim \pi} \mathbb{E}_t \mathbb{E}_{\zeta \sim \mathcal{N}(0, I_{d_y})} \left[ (\zeta_k - U_k(h(\zeta, t, \boldsymbol{y}), t, \boldsymbol{x}))^2 \right].$$

With Theorem 1, we have $S^*(\boldsymbol{y}, t, \boldsymbol{x}) = \nabla_{\boldsymbol{y}} \log p_{\boldsymbol{x},t}(\boldsymbol{y})$ almost everywhere, with $S^*$ defined as

$$S^* \in \underset{S \in \mathcal{S}^+}{\arg \min} \, \mathbb{E}_{\pi(\boldsymbol{x})} \mathbb{E}_t \mathbb{E}_{\pi(\boldsymbol{y}_{\boldsymbol{x}}(0))} \mathbb{E}_{p_{\boldsymbol{x},0t}} \left[ \left\| \nabla_{\boldsymbol{y}_{\boldsymbol{x}}(t)} \log p_{0t}(\boldsymbol{y}_{\boldsymbol{x}}(t) \mid \boldsymbol{y}_{\boldsymbol{x}}(0)) - S(\boldsymbol{y}_{\boldsymbol{x}}(t), t, \boldsymbol{x}) \right\|^2 \right].$$

We have:

$$\mathbb{E}_{\pi(\boldsymbol{x})} \mathbb{E}_t \mathbb{E}_{\pi(\boldsymbol{y}_{\boldsymbol{x}}(0))} \mathbb{E}_{p_{\boldsymbol{x},0t}} \left[ \left\| \nabla_{\boldsymbol{y}_{\boldsymbol{x}}(t)} \log p_{0t}(\boldsymbol{y}_{\boldsymbol{x}}(t) \mid \boldsymbol{y}_{\boldsymbol{x}}(0)) - S(\boldsymbol{y}_{\boldsymbol{x}}(t), t, \boldsymbol{x}) \right\|^2 \right] \tag{19}$$

$$= \mathbb{E}_{\pi(\boldsymbol{x})} \mathbb{E}_t \mathbb{E}_{\pi(\boldsymbol{y}_{\boldsymbol{x}}(0))} \mathbb{E}_{p_{\boldsymbol{x},0t}} \left[ \left\| \frac{m(t; \boldsymbol{y}_{\boldsymbol{x}}(0)) - \boldsymbol{y}_{\boldsymbol{x}}(t)}{\sigma(t)^2} - S(\boldsymbol{y}_{\boldsymbol{x}}(t), t, \boldsymbol{x}) \right\|^2 \right] \tag{20}$$

$$= \sum_{k=1}^{d_y} \mathbb{E}_{\pi(\boldsymbol{x})} \mathbb{E}_t \mathbb{E}_{\pi(\boldsymbol{y}_{\boldsymbol{x}}(0))} \mathbb{E}_{p_{\boldsymbol{x},0t}} \left[ \frac{1}{\sigma(t)} \left( \frac{m(t; \boldsymbol{y}_{\boldsymbol{x}}(0))_k - \boldsymbol{y}_{\boldsymbol{x}}(t)_k}{\sigma(t)} - \sigma(t) S(\boldsymbol{y}_{\boldsymbol{x}}(t), t, \boldsymbol{x})_k \right) \right]^2 \tag{21}$$

$$= \sum_{k=1}^{d_y} \mathbb{E}_{\pi(\boldsymbol{x},\boldsymbol{y})} \mathbb{E}_t \mathbb{E}_{\zeta \sim \mathcal{N}(0, I_{d_y})} \left[ \frac{1}{\sigma(t)} \left( -\zeta_k - \sigma(t) S(\boldsymbol{y} + \sigma(t)\zeta, t, \boldsymbol{x})_k \right) \right]^2 \tag{22}$$

where we used the following facts:

- from Eq. (19) to Eq. (20): the closed-form expression of the score $S$ from Eq. (17),

- from Eq. (20) to Eq. (21): expanding the norm and switching the expectations with the finite sum over $k$,

- from Eq. (21) to Eq. (22): reparametrizing the expectation of $\boldsymbol{y}_{\boldsymbol{x}}(t) \mid \boldsymbol{y}_{\boldsymbol{x}}(0)$ which by definition is a normal distribution with mean $m(t; \boldsymbol{y}_{\boldsymbol{x}}(0))$ and variance $\sigma(t)^2$.

The final manipulation is to remember that theorem 1 and the theorems from Vincent [19] are valid with expectations $\mathbb{E}_t$ against any strictly positive measure of $t$ over $[0, T]$. In particular, we can absorb $\frac{1}{\sigma(t)}$ as a reweighted non-negative measure, and we obtain exactly the definition of $U_k^*$ by defining $U(\boldsymbol{y}, t, \boldsymbol{x}) = \sigma(t) S(\boldsymbol{y} + \sigma(t)\zeta, t, \boldsymbol{x})$ which concludes the proof. $\qquad\square$

## C  Treeffuser default configuration

We use the following configuration as defaults for Treeffuser.

**Forward diffusion.**  We use the variance exploding SDE [11] for all experiments, defined by setting:

$$f(\boldsymbol{y}, t) = 0 \quad \text{and} \quad g(t) = \sqrt{\frac{\mathrm{d}[\sigma(t)^2]}{\mathrm{d}t}}, \tag{23}$$

where $\sigma$ is a given increasing function defined by,

$$\sigma(t) = \alpha_{\min} \left( \frac{\alpha_{\max}}{\alpha_{\min}} \right)^t \tag{24}$$

and $\alpha_{\min} = 0.01$ and $\alpha_{\max} = 20$. For all our experiments we let $t \in [0, 1]$ (i.e. $T = 1$).

**Gradient-boosted tree (GBT) parameters and dataset repetitions.**  We provide a short description of each hyper-parameter of the model alongside the default value. Treeffuser uses LightGBM [22] to learn the GBTs.

- `n estimators` (3000): Specifies the maximum number of trees that will be fit, regardless of whether the stopping criterion is met.
- `learning rate` (0.1): Specifies the shrinkage to use for every tree.
- `num leaves` (31): Specifies the maximum number of leaves a tree can have.
- `early stopping rounds` (50): Specifies how long to wait without a validation loss improvement before stopping.
- `n repeats` (30): Specifies how many Monte Carlo samples to draw per data point to estimate $\mathbb{E}_{t,\zeta}$ in equation Eq. (9).

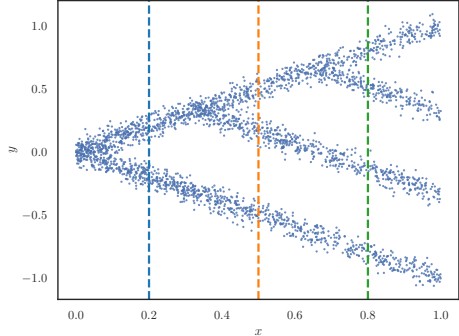
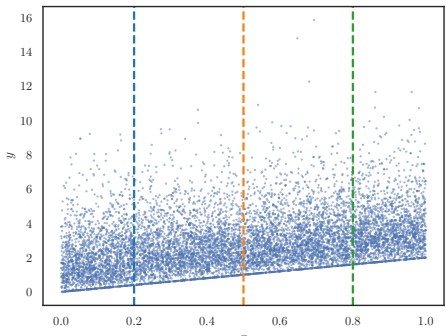

| (a) Simulation 1: branching Gaussian mixture. | (b) Simulation 2: shifted and inflated Gamma. |

Figure 4: Visualization of ground-truth samples for the one-dimensional synthetic datasets used in the empirical studies.

## D Experiments: description of computer resources

All benchmark tasks presented in Section 5.1 and Section 5.2 were run on a cluster with one job per triplet of (model, dataset, split index).

The real world experiments totalled 1135h for 700 tasks (10 datasets, 7 methods, 10 splits). Each task was allocated 4 cpus.

The synthetic experiments totalled 290 hours, for 800 tasks (16 datasets, 5 methods, 10 splits). Each task was allocated 2 CPU.

The M5 experiments were run on a single MacBook Pro over a couple hours. Other figures such as Fig. 1 were also generated on a MacBook pro in a few seconds.

## E Experiments on synthetic data (supplement)

### E.1 Arbitrarily complex synthetic data

We provide details on the three synthetic data experiments presented in Section 5.1 and illustrated in Fig. 1. Each experiment introduces a different distribution of the response variable $y$ given the covariate $x$. The first experiment generates multimodal responses from a branching Gaussian mixture, the second experiment generates inflated responses from a mixture of a shifted Gamma distribution and an atomic measure, and the third experiment generates two-dimensional responses from a nonlinear multioutput regression.

We provide additional visualization of samples for the one-dimensional datasets in Fig. 4.

**Models** We assume that $x \sim \mathrm{Uniform}([0, 1])$ and model the conditional distribution of the response $y$ given $x$ differently for each of the synthetic dataset. Scatter plots of synthetic data from these distributions are provided in Fig. 4

*Branching Gaussian mixture.* We generate multimodal responses from a mixture of equally-weighted Gaussians. In our experiments, we fix the scale $\sigma = 0.05$ and let the number of components and their means scale with $x$ as follows:

$$y \mid x \sim \begin{cases} \mathrm{GaussianMixture}_\sigma(x, -x), & 0 \le x \le 1/3; \\ \mathrm{GaussianMixture}_\sigma(x, 2/3 - x, -x), & 1/3 < x \le 2/3; \\ \mathrm{GaussianMixture}_\sigma(x, 4/3 - x, 2/3 - x, -x), & 2/3 \le x \le 1; \end{cases} \quad (25)$$

where $\mathrm{GaussianMixture}_\sigma(\mu_1, \mu_2, \dots, \mu_K)$ denotes a Gaussian mixture distribution with $K$ equally-weighted components, each with mean $\mu_k$ and scale $\sigma$.

*Covariate inflated Gamma.* We generate inflated responses through the following mixture of a shifted Gamma distribution and an atomic measure:

$$z \sim \text{Bernoulli}(p), \qquad y \mid x, z \sim \begin{cases} \text{ShiftedGamma}(k, \theta, x), & z = 0; \\ \delta_x, & z = 1; \end{cases} \tag{26}$$

where $\text{ShiftedGamma}(k, \theta, x)$ denotes a Gamma distribution with scale $k$ and shape $\theta$ shifted by $x$. In words, given $x$, we set $y$ to be equal to $x$ with probability $p$, and to be drawn from a Gamma shifted by $x$ otherwise. Hence, the conditional distribution of $y$ given $x$ is $x$-inflated and its support $[x, \infty)$ changes with $x$. In our experiments, we set $p = 0.15$, $k = 2$, $\theta = 1$.

*Nonlinear multioutput regression.* We generate two-dimensional responses from a density $p(\boldsymbol{y} \mid x) \propto \exp\left(-\frac{\text{dist}(\boldsymbol{y}, C(x))^2}{2\sigma^2}\right)$ where $\sigma = 0.05$, $\text{dist}(a, B)$ measures the shortest Euclidean distance from $a \in \mathbb{R}$ to the set $B \subset \mathbb{R}^2$, and $C(x)$ is a circular arc centered at $(0, 0)$ with radius $\text{radius}(x)$ and angles of extremities $\theta_1, \theta_2$ evolving as functions of $x$. Intuitively, they interpolate the circular arc represented in Section 1.

The functions are defined as follows.

- If $x \leq 0.5$
    - $\text{radius}(x) = \ell(x) \cdot 1 + r(x) \cdot 0.1$,
    - $\theta_0(x) = \ell(x) \cdot (-0.05) + r(x) \cdot (-0.375)$,
    - $\theta_1(x) = \ell(x) \cdot 0.3 + r(x) \cdot 0.625$,
  where $\ell(x) = (0.5 - x)/(0.5 - 0.17)$, $r(x) = 1 - \ell(x)$.
- If $x > 0.5$
    - $\text{radius}(x) = \ell(x) \cdot 0.1 + r(x) \cdot 1$,
    - $\theta_0(x) = \ell(x) \cdot 0.125 + r(x) \cdot 0.45$,
    - $\theta_1(x) = \ell(x) \cdot 1.125 + r(x) \cdot 0.8$,
  where $\ell(x) = (0.83 - x)/(0.83 - 0.5)$, $r(x) = 1 - \ell(x)$.

### E.2 Data with normal likelihood

We study the performance of Treeffuser in a setting where a simpler model is better adjusted to the generating process. In particular, we let $N$ be number of data points sampled and $d_x$ the dimension and use a linear generative model

$$\boldsymbol{\beta} \sim \mathcal{N}(0, I_{d_x})$$
$$\boldsymbol{x}_i \sim \mathcal{N}(0, I_{d_x}) \quad i \in [N]$$
$$\boldsymbol{y_i} \sim \mathcal{N}(10\boldsymbol{\beta}^\top \boldsymbol{x}, \sigma^2) \quad i \in [N]$$

where $\sigma^2 = 1$. The results are provided in table 3 using the same configuration described in section section 5.2.

**Results** From Table 3, we observe that while Treeffuser remains competitive, Deep Ensembles and NGBoost now demonstrate either equal or very close performance. This is expected since both models assume a normal distribution for the outcomes and the model assumed by Deep Ensembles closely resembles the real generative model. Interestingly, Treeffuser outperforms both iBUG and quantile regression (QReg), suggesting that its superior performance is not merely due to using LightGBM or XGBoost. Finally, we note that Treeffuser continues to perform well in low-data situations compared to other models.

## F  Experiments on standard datasets (supplement)

### F.1  Baseline methods.

We briefly describe the methods used as baselines in our empirical studies.

| $N$ | $d_x$ | Deep Ens. (oracle) | IBUG (oracle) | NGBoost (oracle) | QReg | Treeffuser | |
|---|---|---|---|---|---|---|---|
| 100 | 1 | **2.62**±1.67 | 8.65±10.30 | 4.08±4.35 | 5.00±4.59 | **4.01**±4.11 | $\times 10^{-2}$ |
| 100 | 5 | **3.72**±1.60 | 4.45±1.67 | 4.63±1.75 | 4.69±2.02 | **3.86**±1.67 | $\times 10^{-1}$ |
| 100 | 10 | 4.04±1.77 | 5.31±2.23 | **4.58**±1.60 | 5.13±2.20 | 4.90±2.08 | $\times 10^{-1}$ |
| 100 | 20 | 4.27±2.65 | 5.15±2.29 | 4.49±2.30 | 5.87±3.03 | 4.63±2.19 | $\times 10^{-1}$ |
| 500 | 1 | **0.84**±0.41 | 1.58±0.66 | **0.87**±0.41 | 1.36±0.73 | 1.11±0.44 | $\times 10^{-2}$ |
| 500 | 5 | **3.19**±0.27 | 4.00±0.55 | **3.82**±0.58 | 4.44±0.74 | 3.85±0.51 | $\times 10^{-1}$ |
| 500 | 10 | **3.44**±0.71 | 4.25±0.87 | 4.03±0.90 | 4.51±1.10 | **3.98**±0.83 | $\times 10^{-1}$ |
| 500 | 20 | 3.69±0.81 | **4.31**±1.04 | 4.82±1.50 | 4.80±1.18 | 4.32±0.91 | $\times 10^{-1}$ |
| 1000 | 1 | **2.51**±1.08 | 10.59±2.57 | **7.10**±2.09 | 11.85±2.34 | 9.72±2.47 | $\times 10^{-3}$ |
| 1000 | 5 | **3.18**±0.26 | 3.67±0.39 | 3.73±0.33 | 3.66±0.34 | **3.52**±0.21 | $\times 10^{-1}$ |
| 1000 | 10 | **3.29**±0.42 | 4.01±0.63 | 4.01±0.54 | 4.23±0.76 | **3.95**±0.59 | $\times 10^{-1}$ |
| 1000 | 20 | **3.40**±0.29 | 4.25±0.34 | 4.11±0.47 | 4.27±0.49 | **4.00**±0.38 | $\times 10^{-1}$ |
| 5000 | 1 | **0.56**±0.25 | 9.25±2.71 | **3.87**±2.26 | 10.36±2.96 | 10.00±2.49 | $\times 10^{-3}$ |
| 5000 | 5 | **3.27**±0.26 | 3.69±0.36 | 3.63±0.33 | 3.63±0.34 | **3.63**±0.29 | $\times 10^{-1}$ |
| 5000 | 10 | **3.18**±0.18 | 3.82±0.26 | 3.66±0.25 | 3.65±0.27 | **3.59**±0.25 | $\times 10^{-1}$ |
| 5000 | 20 | **3.21**±0.20 | 3.86±0.23 | 3.90±0.25 | 3.80±0.24 | **3.69**±0.22 | $\times 10^{-1}$ |

Table 3: CRPS (lower is better) by dataset and method. Treeffuser provides comparable predictions to methods with parametric assumptions that match the generating process. The standard deviations are measured with 10-fold cross-validation.

*NGBoost (parametric, tree-based).* NGBoost [14] models the conditional distribution of the target variable using a parametric family of distributions whose parameters are predicted by a gradient-boosting algorithm. It uses natural gradients for more stable, accurate, and faster learning. In our experiments, we set the parametric model to be Gaussian.

*Deep ensembles (parametric, nnet-based).* A Deep ensemble [17] is a collection of neural networks that are individually trained to model a parametric conditional distribution $p(y|x)$. The neural networks are then combined to obtain a mixture. We set the parametric model to be Gaussian.

*Quantile regression (nonparametric, tree-based).* Quantile regression [60] estimates the quantiles of the conditional distribution $p(y|x)$. We implemented a GBT-based version of the method by fitting trees with inputs $\boldsymbol{x}$ and $q$, where $q$ is the probability of the desired quantile. During training, we optimized the objective $\mathbb{E}_q \mathbb{E}_{\boldsymbol{x},\boldsymbol{y}}[L(Q(\boldsymbol{x},\boldsymbol{y}),q)]$, where $L$ is the pinball loss and $q$ is sampled from a zero-one uniform distribution. In our experiments, we used LightGBM as a GBT method.

*IBUG (parametric, tree-based).* IBUG [15] extends any GBT into a probabilistic estimator. Given an input instance $x$, it outputs a distribution around the prediction using the $k$-nearest training data points. The distance between a training instance $x_0$ and $x$ depends on the number of co-occurrences of $x_0$ and $x$ across the leaves of the ensemble. In our experiments, we used XGBoost and a Gaussian likelihood, following the default specifications of the method.

*DRF (nonparametric, tree-based).* DRF [50] grows a random forest by maximizing the differences in the response distributions across split groups with a distributional metric. Repeated randomization induces a weighting function that assesses how relevant a training data point is to a given input $x$. These weights are used to return a weighted empirical distribution of the response.

## F.2 Hyperparameters of the methods.

Unless specified otherwise, we tuned the hyperparameters of each method using Bayesian optimization for 25 iterations. We used the following search space for each method (where $[\![a, b]\!]$ denotes the set of integers from $a$ to $b$):

- Treeffuser
    - `n estimators` $\in [\![100, 3000]\!]$
    - `n repeats` $\in [\![10, 50]\!]$
    - `learning rate` $\in [0.01, 1]$ (log-uniform)
    - `early stopping rounds` $\in [\![10, 100]\!]$
    - `num leaves` $\in [\![10, 100]\!]$

- Quantile regression (QReg)
  - `n estimators` $\in [\![100, 3000]\!]$ (log-uniform)
  - `n repeats` $\in [\![10, 100]\!]$
  - `learning rate` $\in [0.01, 1]$
  - `early stopping rounds` $\in [\![10, 100]\!]$
  - `num leaves` $\in [\![10, 100]\!]$
- IBUG
  - `k` $\in [\![20, 250]\!]$
  - `n estimators` $\in [\![10, 1000]\!]$
  - `learning rate` $\in [0.01, 0.5]$ (log-uniform)
  - `max depth` $\in [\![1, 100]\!]$
- DRF
  - `min node size` $\in [\![5, 30]\!]$
  - `num trees` $\in [\![250, 3000]\!]$
- NGBOOST
  - `n estimators` $\in [\![100, 10000]\!]$
  - `learning rate` $\in [0.005, 0.2]$
- Deep ensemble
  - `n layers` $\in [\![1, 5]\!]$
  - `hidden size` $\in [\![10, 500]\!]$
  - `learning rate` $\in [10^{-5}, 10^{-2}]$ (log-uniform)
  - `n ensembles` $\in [\![2, 10]\!]$

For the methods that only return point predictions, we used the following search spaces:

- XGBoost
  - `n estimators` $\in [\![10, 1000]\!]$
  - `learning rate` $\in [0.01, 0.5]$ (log-uniform)
  - `max depth` $\in [\![1, 100]\!]$
- LightGBM
  - `n estimators` $\in [\![10, 1000]\!]$
  - `learning rate` $\in [0.01, 0.5]$ (log-uniform)
  - `num leaves` $\in [\![10, 100]\!]$
  - `early stopping rounds` $\in [\![10, 100]\!]$

## F.3 Accuracy and calibration results.

| dataset | $N, d_x, d_y$ | Deep ensembles | NGBoost (Gaussian) | iBUG (XGBoost) | Quantile regression | DRF | Treeffuser | Treeffuser (no tuning) | |
|---|---|---|---|---|---|---|---|---|---|
| bike | 17379,12,1 | **3.70**±**0.13** | 11.52±0.30 | 4.11±0.19 | 4.63±0.21 | 4.86±0.18 | **3.69**±**0.14** | 3.81±0.15 | ×10¹ |
| energy | 768, 8, 2 | 11.34±2.08 | 11.41±1.25 | NA | NA | 13.73±1.69 | **8.32**±**1.35** | **8.79**±**1.46** | ×10⁻¹ |
| kin8nm | 8192, 8, 1 | **0.64**±**0.02** | 1.71±0.06 | 1.38±0.08 | 1.18±0.04 | 1.72±0.04 | **1.06**±**0.03** | 1.06±0.02 | ×10⁻¹ |
| movies | 7415, 9, 1 | 9.35±1.90 | × | 9.83±1.44 | 18.23±2.61 | 15.87±2.59 | **8.85**±**1.66** | **8.63**±**1.38** | ×10⁷ |
| naval | 11934,17,1 | 10.42±1.69 | 8.66±0.61 | 6.46±1.07 | 148.80±35.50 | 11.49±0.98 | **4.40**±**0.26** | **4.92**±**0.57** | ×10⁻⁴ |
| news | 39644,58,1 | 1.95±2.66 | × | 1.18±0.40 | 1.12±0.40 | **1.08**±**0.41** | 1.10±0.40 | **1.09**±**0.41** | ×10⁴ |
| power | 9568, 4, 1 | 3.81±0.32 | 3.66±0.35 | 3.06±0.32 | 36.27±0.56 | 3.65±0.35 | **2.93**±**0.30** | **3.02**±**0.30** | ×10⁰ |
| superc. | 21263,81,1 | 11.26±0.74 | 11.25±0.87 | 9.67±0.54 | 9.36±0.48 | 10.79±0.56 | **9.08**±**0.48** | **9.21**±**0.53** | ×10⁰ |
| wine | 6497, 12, 1 | 6.54±0.17 | 6.85±0.18 | 6.02±0.26 | 6.28±0.29 | 6.84±0.21 | **5.88**±**0.17** | **5.98**±**0.19** | ×10⁻¹ |
| yacht | 308, 6, 1 | 1.06±0.46 | **0.83**±**0.31** | **0.83**±**0.31** | 1.22±0.74 | 2.30±1.07 | 2.10±1.08 | 2.08±0.45 | ×10⁻¹ |

Table 4: RMSE (lower is better) by dataset and method. × indicates the method failed to run, and NA that the method is not directly applicable to multivariate outputs. The standard deviations are measured with 10-fold cross-validation. For each dataset, the two best methods are bolded. Treeffuser provides the most accurate probabilistic predictions, even when initialized with defaults.

| dataset | $N, d_x, d_y$ | Deep ensembles | NGBoost (Gaussian) | iBUG (XGBoost) | Quantile regression | DRF | Treeffuser | Treeffuser (no tuning) | |
|---|---|---|---|---|---|---|---|---|---|
| bike | 17379,12,1 | 3.28±0.97 | 17.79±0.68 | 5.77±1.04 | 2.55±0.29 | 3.45±0.43 | **1.73**±**0.62** | **1.86**±**0.87** | ×10⁻² |
| energy | 768, 8, 2 | 8.08±1.41 | 5.82±1.74 | NA | NA | **4.54**±**0.74** | **4.11**±**1.14** | 5.53±1.42 | ×10⁻² |
| kin8nm | 8192, 8, 1 | **2.58**±**1.50** | 4.70±0.44 | 2.99±0.58 | 8.47±0.80 | 2.79±0.52 | 3.85±0.96 | **3.34**±**1.22** | ×10⁻² |
| movies | 7415, 9, 1 | 5.75±1.38 | × | 7.56±3.31 | 49.93±0.07 | **1.31**±**0.47** | **2.71**±**1.36** | 5.17±1.32 | ×10⁻² |
| naval | 11934,17,1 | 13.31±1.90 | **5.50**±**0.75** | **3.86**±**2.59** | 15.05±0.58 | 17.37±0.40 | 7.75±0.41 | 9.10±0.39 | ×10⁻² |
| news | 39644,58,1 | 11.36±0.63 | × | 14.43±1.82 | 18.96±0.42 | **1.14**±**0.25** | 3.50±0.70 | **4.87**±**0.28** | ×10⁻² |
| power | 9568, 4, 1 | 2.52±1.18 | 2.53±1.00 | 3.67±1.42 | 4.75±0.93 | **2.37**±**0.58** | 4.36±1.31 | **1.77**±**0.74** | ×10⁻² |
| superc. | 21263,81,1 | 3.19±0.63 | 2.75±0.69 | 5.39±1.97 | 5.47±0.44 | 3.87±0.32 | **1.34**±**0.10** | **1.33**±**0.16** | ×10⁻² |
| wine | 6497, 12, 1 | **1.92**±**0.56** | **2.68**±**0.55** | 4.72±0.83 | 4.36±0.91 | 22.02±1.13 | 5.70±0.55 | 5.22±0.69 | ×10⁻² |
| yacht | 308, 6, 1 | 13.35±2.17 | 9.04±2.41 | 9.06±3.26 | 8.26±2.80 | **7.15**±**2.86** | **7.27**±**2.58** | 10.02±2.29 | ×10⁻² |

Table 5: MACE (lower is better) by dataset and method. × indicates the method failed to run, and NA that the method is not directly applicable to multivariate outputs. The standard deviations are measured with 10-fold cross-validation. For each dataset, the two best methods are bolded. Treeffuser has a competitive calibration error across most datasets.

| dataset | $N, d_x, d_y$ | iBUG (XGBoost) | XGBoost | LightGBM | Treeffuser | Treeffuser (no tuning) | |
|---|---|---|---|---|---|---|---|
| bike | 17379,12,1 | 4.11 ± 0.19 | **3.72 ± 0.13** | 3.79 ± 0.12 | **3.69 ± 0.14** | 3.81 ± 0.15 | ×10¹ |
| kin8nm | 8192, 8, 1 | 1.38 ± 0.08 | 1.12 ± 0.04 | **1.04 ± 0.03** | **1.06 ± 0.03** | 1.06 ± 0.02 | ×10⁻¹ |
| movies | 7415, 9, 1 | 9.83 ± 1.44 | 8.96 ± 1.60 | 9.08 ± 1.76 | **8.85 ± 1.66** | **8.63 ± 1.38** | ×10⁷ |
| naval | 11934,17,1 | 6.46 ± 1.07 | 5.11 ± 0.45 | 6.68 ± 1.53 | **4.40 ± 0.26** | **4.92 ± 0.57** | ×10⁻⁴ |
| news | 39644,58,1 | 1.18 ± 0.40 | **1.09 ± 0.40** | **1.09 ± 0.40** | 1.10 ± 0.40 | 1.09 ± 0.41 | ×10⁴ |
| power | 9568, 4, 1 | 3.06 ± 0.32 | **2.97 ± 0.33** | 2.99 ± 0.32 | **2.93 ± 0.30** | 3.02 ± 0.30 | ×10⁰ |
| superc. | 21263,81,1 | 9.67 ± 0.54 | **9.04 ± 0.48** | 9.21 ± 0.44 | **9.08 ± 0.48** | 9.21 ± 0.53 | ×10⁰ |
| wine | 6497, 12, 1 | 6.02 ± 0.26 | 5.95 ± 0.23 | **5.90 ± 0.27** | **5.88 ± 0.17** | 5.98 ± 0.19 | ×10⁻¹ |
| yacht | 308, 6, 1 | **0.83 ± 0.31** | 2.19 ± 1.82 | **0.67 ± 0.40** | 2.10 ± 1.08 | 2.08 ± 0.45 | ×10⁻¹ |

Table 6: RMSE (lower is better) by dataset and method. This table extends Table 4 by includes vanilla XGBoost and LightGBM, alongside iBUG and Treeffuser. The standard deviations are measured with 10-fold cross-validation. For each dataset, the two best methods are bolded.

## G  Ablation study on noise scaling

We evaluate the impact of including noise scaling in the score parametrization of Eq. (13), specifically $S(\boldsymbol{y}, t, \boldsymbol{x}) = U(\boldsymbol{y}, t, \boldsymbol{x})/\sigma(t)$, where $S(\boldsymbol{y}, t, \boldsymbol{x})$ denotes the score function of $\boldsymbol{y}(t) \mid \boldsymbol{x}$, $U$ the gradient-boosted trees, and $\sigma(t)$ the noise schedule of the diffusion process. We compare Treeffuser with and without noise scaling under the same experimental setup of Section 5.2.

Table 7 presents the CRPS by dataset and method. Treeffuser without noise scaling fails to achieve any reasonable CRPS. We believe Treeffuser without noise scaling may require extra model capacity (e.g., deeper trees and more training iterations) to handle the varying noise levels adequately. However, we did not further optimize it, as noise scaling provides state-of-the-art results without the need for additional complexity.

| Treeffuser | bike | energy | kin8nm | naval | news | power | superc. | wine | yacht |
|---|---|---|---|---|---|---|---|---|---|
| with scaling | $1.60_{\pm 0.05}$ | $3.07_{\pm 0.40}$ | $5.89_{\pm 0.14}$ | $2.02_{\pm 0.08}$ | $1.98_{\pm 0.17}$ | $1.49_{\pm 0.07}$ | $3.52_{\pm 0.13}$ | $2.59_{\pm 0.13}$ | $3.11_{\pm 0.99}$ |
| without scaling | $176_{\pm 211}$ $\times 10^1$ | $5258_{\pm 9667}$ $\times 10^{-1}$ | $204_{\pm 240}$ $\times 10^{-2}$ | $154_{\pm 175}$ $\times 10^{-4}$ | $62.1_{\pm 78.4}$ $\times 10^3$ | $289_{\pm 296}$ $\times 10^0$ | $112_{\pm 162}$ $\times 10^0$ | $30_{\pm 53}$ $\times 10^{-1}$ | $18682_{\pm 41026}$ $\times 10^{-1}$ |

Table 7: CRPS (lower is better) by dataset for Treeffuser with and without the noise scaling reparametrization in Eq. (13). Without scaling, Treeffuser does not produce meaningful results.

# H  Runtime and Complexity

| | m5 | movies | bike | energy | kin8nm | naval | news | power | superc. | wine | yacht |
|---|---|---|---|---|---|---|---|---|---|---|---|
| $\frac{\text{Time}}{\text{sample}}$ | 1.33 | 0.68 | 1.33 | 0.55 | 1.23 | 1.46 | 1.31 | 1.15 | 0.85 | 0.56 | 0.28 |

Table 8: Average running time in milliseconds for producing a sample from the model.

To demonstrate the efficiency of Treeffuser, we present the results of the following experiments and benchmarks:

1. We run Treeffuser on subsets of the M5 dataset (section 5.3) of various sizes and report the training time in Fig. 5.

2. We report the training time for running Treeffuser on the datasets used in the paper in Fig. 6.

3. We report in Table 8 the average runtime for generating a single sample after training Treeffuser. The average is calculated by sampling five thousand points after training the model with default settings and 50 discretization steps.

We conducted all of these experiments on a 2020 MacBook Pro with a 2.6 GHz 6-Core Intel Core i7 processor.

From these results we find that:

- Treeffuser is very fast at fitting moderately sized datasets, with runtime increasing linearly with dataset size.

- Sampling a single points is very fast ($\approx 10^{-3}$ seconds) yet, drawing many samples can become significant, e.g., with a large test set.

With respect to the time complexity of the model we note the following. If the time complexity of fitting a single GBT is of order $O(F(|\mathcal{D}|))$, where $F : \mathbb{R}_+ \to \mathbb{R}_+$ and $|\mathcal{D}|$ is the size of the dataset, then the time complexity of fitting Treeffuser is $O(d \times F(\texttt{n\_repeats} \times |\mathcal{D}|))$, where $d$ is the dimension of $\boldsymbol{y}$ and $\texttt{n\_repeats}$ is the parameter that determines how many noisy versions of the dataset are sampled. Similarly, assuming constant time per GBT evaluation, the complexity of sampling is $O(\texttt{n\_samples} \times \texttt{n\_discretization\_steps})$, where $\texttt{n\_samples}$ is the number of samples drawn from the model, and $\texttt{n\_discretization\_steps}$ is the number of function evaluations used by the SDE solver.

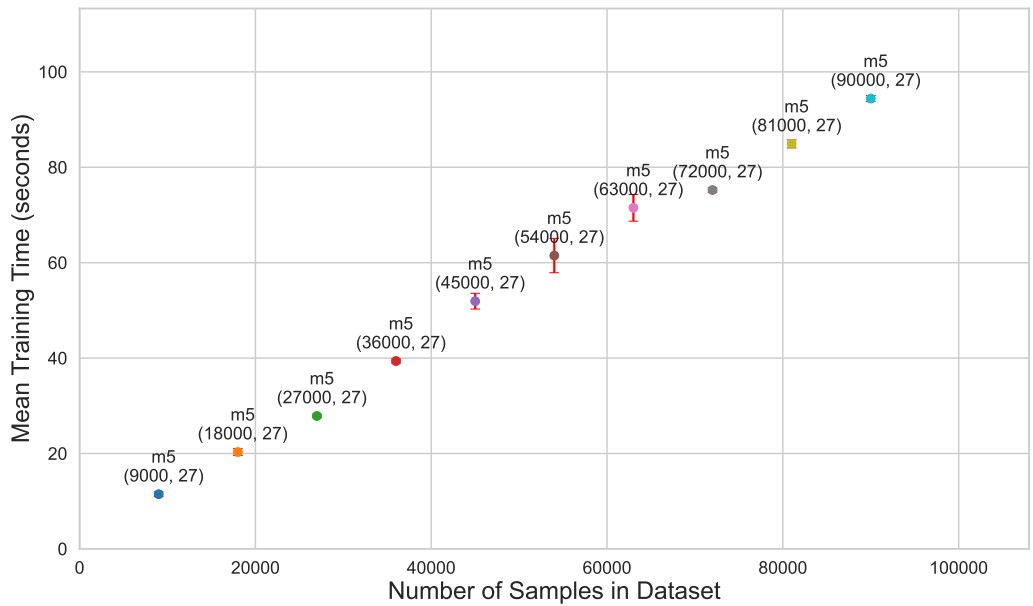

Figure 5: Dataset size vs training time on subsets of the M5 dataset. Error bars are computed over 5 runs. Treeffuser training speed grows linearly with the size of the training points.

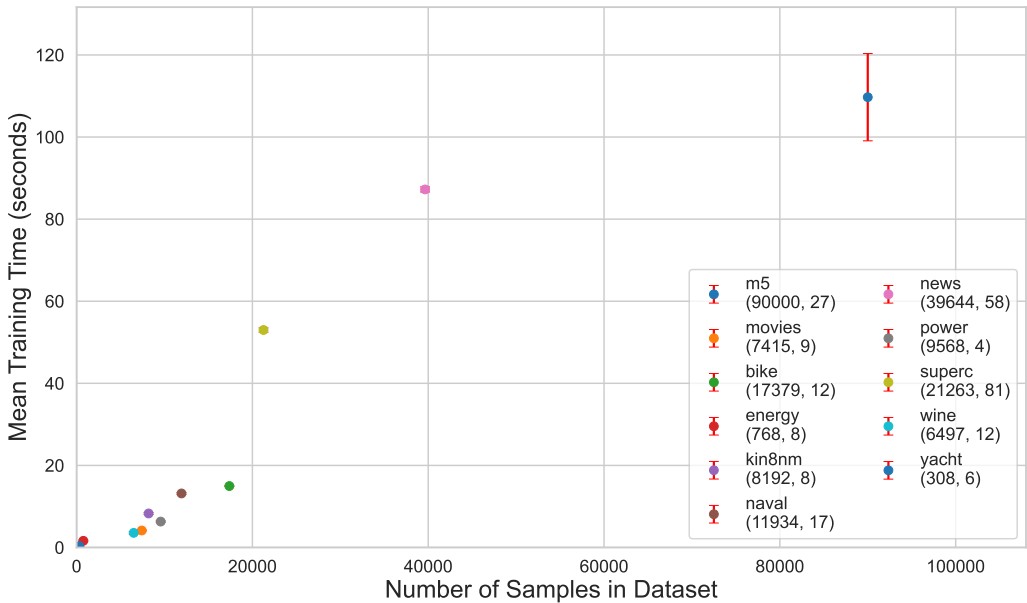

Figure 6: Dataset size vs training time on benchmark datasets. Error bars are computed over 5 runs. Treeffuser trains in under 120 seconds for all datasets.

