# OpenReview forum: "Treeffuser: probabilistic prediction via conditional diffusions with gradient-boosted trees"
_NeurIPS.cc/2024/Conference — NeurIPS 2024 poster_

### Official Review · Reviewer_knMV · 2024-06-24

**Soundness:** 3
**Presentation:** 3
**Contribution:** 4
**Rating:** 6
**Confidence:** 4

**Summary:**

The paper proposes Treeffuser, a nonparametric method for modeling the output distribution. Treeffuser learns a diffusion model using gradient-boosted trees, and uses conditional diffusion models to produce a distribution over the response variable given an input vector. Since Treeffuser user gradient-boosted trees, it is adept at providing accurate predictive distributions for tabular data problems. Experiments on toy examples, real-world datasets, and a case study highlight the flexibility and effectiveness of Treeffuser for accurately modeling the output distribution for a wide range of use cases.

**Strengths:**

- The experimental results section demonstrates Treeffuser's effectiveness on complex contrived examples, real-world tabular regression datasets in comparison with existing methods, and a realistic case study.

- The paper is well-written and relatively easy to follow.

- The proposed approach is very flexible and nonparametric, enabling it to model any type of response distribution. This aspect is especially valuable for tabular regression problems, in which GBDTs do not inherently provide any type of uncertainty in their predictions.

- Treeffuser works well even with default hyperparameters.

**Weaknesses:**

- As the authors note, Treeffuser must solve a stochastic differential equation to generate samples, which can become expensive if there are a large number of example to predict and each prediction requires the generation of a large number of samples.

- No complexity or empirical runtime analysis is provided. This type of analysis would be greatly beneficial to readers and practitioners to help them decide if and when to use Treeffuser for their particular problem.

- The comparison to other methods (Section 5.2) could be improved. For example, iBUG's $k$ hyperparameter is tuned using values [20, 150]; however, $k$ is a critical hyperparameter, and increasing the number of potential values may significantly increase iBUG's performance. Additionally, iBUG can non-parametrically model the response using kernel density estimation (KDE) which may improve its performance on datasets in which the output is not expected to be Gaussian.

**Questions:**

- Do the authors have negative log likelihood (NLL) results?

- I'm surprised Treeffuser performs better than iBUG for point predictions (Table 4) since iBUG is simply using the underlying GBDT model, do the authors have a hypothesis as to why Treeffuser performs better than XGBoost in terms of point performance?

**Limitations:**

Yes, the authors address the limitations of their work.

---

> ### Author Rebuttal · Authors · 2024-08-06
>
> # Rebuttal Reviewer knMV
>
> We thank knMV for their review. We are encouraged that the reviewer found the paper well-written and easy to follow and that the author found our approach very flexible and valuable for tabular regression problems with UQ. Below, we provide additional results and discussion that answer the questions and concerns that the reviewer presented. The new experiments definitely strengthened the paper!
>
> ## Weaknesses
> **W1. Treeffuser must solve a stochastic differential equation to generate samples, which can become expensive**
> **W2. No complexity or empirical runtime analysis provided**
>
> Thank you for the great comments. In the paper, we provided some runtime analysis of Treeffuser lines 257-261. Yet, we agree that more runtime analyses are helpful for readers and practitioners. So we generated three new figures running Treeffuser with default parameters on a 2020 MacBook Pro.
> 1. [Figure 1](https://imgur.com/a/Szo5ihU): runtime in seconds for training Treeffuser on subsets of different sizes of the M5 dataset (section 5.3). The shape of the training data is indicated on the plot. Error bars represent the standard error over five runs of Treeffuser.
> 2. [Figure 2](https://imgur.com/a/CzsPy1Z): runtime in seconds for training Treeffuser on the other datasets used in the paper. Error bars represent the standard error of over five runs of Treeffuser.
> 3. [Figure 3](https://imgur.com/a/5RQbM77): average runtime in seconds for producing one sample after training Treeffuser. The average is computed by sampling five thousand points from Treeffuser using its default settings of 50 discretization steps.
>
> From these results, we find that:
> - Treeffuser is very fast at fitting moderately sized datasets, with runtime increasing linearly with dataset size;
> - sampling a single point is very fast ($\approx 10^{-4}$ seconds) and can be parallelized;
> - yet, drawing many samples can take more time, e.g., with a large test set.
>
> We also agree with the suggestion of including a complexity analysis and will add a detailed discussion in the appendix. To be brief here, the complexity for training is the same as for LightGBM with the subtelty that the dataset size is multiplied by `n_repeats` (see Alg 1 and Appendix C), and a different model is trained for each dimension of $y$. The complexity for sampling is the same as evaluating the trees `n_sampling_steps * y_dim` times.
>
>
>
> **W.3 The comparison to other methods (Section 5.2) could be improved. For example, iBUG's hyperparameter [...] which may improve its performance on datasets in which the output is not Gaussian.**
>
> Thank you for your suggestion. Following your advice, we
> 1. increased the search space of $k$ using the method for optimal tuning of $k$ (Algorithm 3 in iBUG's paper)
> 2. Implemented iBUG with KDE likelihood -- by tuning the kde bandwidth in [0.0001, 1000].
>
> The results are available [here](https://imgur.com/a/D1x4k9S).
>
> The conclusions are unchanged in both cases: Treeffuser outperforms iBUG (with and without KDE). Thank you for suggesting these great baselines; we have updated section 5.2 of our paper with them. We hope that this addresses your concerns.
>
> ## Questions
>
> **Q1. Do the authors have negative log likelihood (NLL) results?**
>
> We thank the reviewer for this question. We do not provide NLL results. While NLL can be approximated using KDEs on the samples outputted by Treeffuser, we found that KDEs are sensitive to the choice of bandwidth. We also note that exact likelihood computations are possible for diffusion models when the score estimator is continuous everywhere, as with neural-based diffusions (see section D.2 [8]). However, this is not applicable for GBT, because trees are not continuous.
>
> Fortunately, this apparent limitation has no practical implications:
> - For evaluation, we use CRPS, which can be evaluated from samples. CRPS is a proper scoring rule [16] with advantages over NLL, such as better handling of tail probabilities [17]. CRPS is commonly used for probabilistic predictions [16,18].
> - For concrete downstream applications, we argue that being able to generate samples from $p(y\mid x)$ is more important than raw values of $\log p(y\mid x)$.
>
>
> **Q2. I'm surprised Treeffuser performs better than iBUG for point predictions [...] Do the authors have a hypothesis as to why Treeffuser performs better than XGBoost in terms of point performance?**
>
> Thank you for your question. We also anticipated that iBug would excel in point prediction since it relies on XGBoost. However, we are not surprised by the results.
>
> 1. First, we nuance that Treeffuser only slightly outperforms iBug for point prediction, with iBug often very close behind.
> 2. Second, we point out that iBug's point predictions do not mathematically coincide with XGBoost's. Indeed, the point predictions of a probabilistic method are computed as the expected mean: the empirical average of 50 samples from the modeled conditional probability distribution $p(y\mid x)$. So if the conditional probability returned by iBUG has errors, then the point prediction might also have errors.
>
> In light of your comment, we conducted additional experiments for point predictions against XGBoost and LightGBM directly (with hyperparameter tuning optimization).
>
> The results are shown in [this table](https://imgur.com/a/4gNpSWd), where we also updated iBUG's point predictions with your suggested hyper-parameter tuning improvements which yielded similar but slightly improved results.
>
> As expected, vanilla XGBoost and LightGBM outperform or tie with all the probabilistic methods. In particular, they are often comparable with Treeffuser, suggesting that Treeffuser provides probabilistic prediction without sacrificing point predictions of the mean.
>
> We updated Table 4 with these results and added further discussion in the paper for clarity. Thank you for raising this point and suggesting the addition of more baselines; they strengthened the paper!

---

> ### Comment · Reviewer_knMV · 2024-08-08
> **Thank you**
>
> I thank the authors for their thorough response, and appreciate the additional experimental results. Overall, my concerns have been largely addressed and I have updated my score accordingly.

---

### Official Review · Reviewer_3Apn · 2024-06-27

**Soundness:** 3
**Presentation:** 3
**Contribution:** 2
**Rating:** 5
**Confidence:** 3

**Summary:**

The authors propose a methodology for computing probabilistic predictions in regression problems using diffusion and gradient-boosted trees. In particular, their methodology can generate samples from p(y | x) from which statistics such as estimated quantiles could be recovered. They find that their method can learn flexible and non-standard distributions well and also works well on real-world datasets.

**Strengths:**

The method results in a tractable standard supervised-ML problem, where the dataset is an augmented version of the original dataset.

The method produces good results on a range of datasets.

The paper goes into detail on the relevant diffusion equations governing the process.

I liked the clever application with the newsvendor problem.

**Weaknesses:**

The work is not clear about its contribution and what parts of the diffusion-based setup are new vs borrowed from existing works.

There could be more and better empirical comparisons.

Notation and clarity could be improved.
* Line 129 defines a vector a but that is not used in equation (11).
* Not obvious how eqn (11) simplifies to eqn (12).

The proposed method may not be easily used or widely applicable because of the complex process needed to generate samples involving solving differential equations.

**Questions:**

I'm left with some basic questions about the method. What part of the contribution is due to the specific way the diffusion process was modeled and what part is due to the use of trees? How does the method differ from other diffusion-based approaches to probabilistic prediction such as the cited CARD paper? What were the key breakthroughs (if there are differences) in how you set up the diffusion problem?

Tagasovska and Lopez-Paz's Single-Model Uncertainties for Deep Learning (NeurIPS 2019) is a relevant work to compare to here. If I understand right it's essentially the same as your quantile regression comparison except you use trees instead of neural networks to learn f(x, q). How would the method perform with neural networks?

How easy is the SDE solver to stitch with the outputs of the trees?

**Limitations:**

Yes

---

> ### Author Rebuttal · Authors · 2024-08-06
>
> # Rebuttal Reviewer 3Apn
>
> We thank Reviewer 3Apn for their valuable feedback. We are pleased to hear that they appreciate our application to the newsvendor problem. Based on their comments and questions, we clarify our contributions, the applicability of our method, and its implementation below.
>
> ## Weaknesses
>
> **W1. The work is not clear about its contribution and what parts of the diffusion-based setup are new vs borrowed from existing works.**
>
> Please see our answer to Q1.
>
> **W2. There could be more and better empirical comparisons.**
>
> Thank you for the comment. We selected five strong approaches that practitioners might choose for probablisitc predictions. However, we look forward to comparing the performance of Treeffuser (our method) against other methods.
>
> What are some specific datasets or methods you think would be valuable for comparison in our paper?
>
> **W3. Notation and clarity could be improved**
>
> Thank you for the feedback.
> - The vector **a** is a dummy vector to explain notations, it is normal to not find it in the equation. We rephrased the sentence to avoid confusion.
> - The derivation of Eq. 12 was shown in Appendix Eqs. 19 to 22. We updated the paper to make sure this is also clear in the main text.
>
> **W4. The proposed method may not be easily used or widely applicable because of the complex process needed to generate samples involving solving differential equations.**
>
> Thank you for raising this point, usability is very important to us. But we are surprised by this comment, because Treeffuser is actually _straightforward to use and widely applicable_. This is a key contribution of our work. We argue this is the case for the following reasons:
>
> - Concretely, we designed a simplified user experience by wrapping Treeffuser in a plug-and-play Python package that solves the differential equations for the user. With the accompanying package in the supplementary material, using Treeffuser is as simple as:
>     ```
>     model = Treeffuser()
>     model.fit(X,y)
>     samples = model.sample()
>     ```
>
> - Offering such a simple user interface is only possible because we built Treeffuser with a flexible estimator (trees) and flexible model (diffusions). As a result, this simple training procedure is the same regardless of the type of data (e.g. heavy tailed, multi-modal, heteroskedastic, categorical, missing).
> - In practice, Treeffuser is accurate on diverse datasets, even without tuning (see our studies in Table 1, 2, and 4).
>
>
> To reiterate, while solving differential equations is indeed complex, it is not more complex than fitting boosted trees or optimizing thousands of neural network parameters. Efficient packages such as `xgboost`, `sklearn`, and `torch` made these tasks accessible to any user. With our `treeffuser` companion package, we aim to make diffusion-based probabilistic predictions for tabular data equally accessible.
>
> ## Questions
>
> **Q1.1 What part of the contribution is due to the specific way the diffusion process was modeled and what part is due to the use of trees?**
>
> Thank you for the question. We think that both parts cannot really be separated. The main contribution and goal is to use trees, but we had to adapt the diffusion model process to do so. (For example, trees only have a 1d output.)
>
> Yet, you are right that the way we modeled the diffusion is important. For example, the score reparametrization on line 133 (and Eq. 13) is crucial for performance.
>
> To highlight the importance of our design choices, we added this discussion to the main text, and included an ablation study on the score reparametrization in the Appendix.
>
> **Q1.2 How does the method differ from other diffusion-based approaches to probabilistic prediction such as the cited CARD paper?**
>
> The differences between CARD[9] and Treeffuser are:
> 1. CARD uses a specialized neural network architecture, while we use out-of-the-box gradient-boosted trees (e.g. XGBoost, LightGBM).
> 2. CARD uses a discrete time approach [13-14], while we formulate our conditional diffusion problem with continuous SDEs [8], which we believe renders the theory easier to follow.
> 3. Our approach requires much fewer function evaluations (i.e. 50 vs 1000) to generate a sample.
> 4. CARD requires two training steps: a first model is trained to predict the mean of the target only, then a second model learns the distribution of the data using a diffusion that takes the prediction from the first model as input. Treeffuser directly trains the diffusion model on the target. It doesn't require training any auxiliary model for conditioning and is thus simpler.
> 5. We attempted to use CARD in our experiments for comparison but were unable to obtain good results.
>
> **Q2. Tagasovska and Lopez-Paz's Single-Model Uncertainties for Deep Learning (NeurIPS 2019) is a relevant work to compare to here. [...] How would the method perform with neural networks?**
>
> Thank you for sharing this reference. We were not aware of it but now include it in the text. Indeed you are correct, the baseline quantile regression used in our paper implements the same idea using trees instead of neural networks.
>
> In fact, we originally tried a similar version of quantile regression with neural networks, and we found that on all our experiments, the tree based quantile regression performed better. This is in line with the observed dominance of GBTs over neural networks on tabular data [10-12].
>
>
> **Q3 How easy is the SDE solver to stitch with the outputs of the trees?**
>
> Integrating the SDE solver with the outputs of the GBT is easy, akin to neural network based diffusions. Concretely, the trees are wrapped into a class with a `predict` function that returns the learned score, which is then fed to the SDE solver.
>
> We ensured our code is modular to support any SDE solver, while hiding all the SDE complexity from the users. For example, `model.sample(n_samples)` calls the standard Euler-Maruyama SDE solver that handles the numerical integration automatically.

---

> > ### Comment · Reviewer_3Apn · 2024-08-12
> >
> > Thank you for your detailed response! I appreciate the authors highlighting ease-of-use of the method as well as additional thoughts about neural networks and GBTs.
> >
> > I'm still a bit unsettled on my question about novelty. My read of the method is that there are two distinct parts of the approach, (1) how to augment the original train set; and (2) what model to fit on the new train set. Presumably one could swap neural networks in for (2), with some impact on quality (perhaps a decrease on simple tabular data but an improvement on complex inputs). That could even be a comparison in this paper to help differentiate between the tree-part of the contribution and the diffusion setup part of the contribution.
> >
> > It's clear to me now that CARD is a different approach to solving the diffusion problem but does that mean the fundamental approach used in this paper (which again, would seemingly work for any model structure) novel? The paper does not clearly say, nor do the authors, which makes me wonder if it is using an existing approach. They say they have adapted diffusion models for trees but without much specificity about what is novel about this adaptation (many models are single-output, not just trees).
> >
> > I will leave my score constant, with the proviso that if the ACs and other reviewers believe there is novelty in the overall approach to solving the diffusion problem (and not just to using trees as step (2) of the solve instead of other model classes), I would support the paper more strongly.

---

> > > ### Author Response · Authors · 2024-08-13
> > >
> > > Thank you for your reply. We believe there might be some misunderstanding regarding the novelty and potential impact of our paper. We would like to clarify it.
> > >
> > > The review appears to overlook the significance of using trees, implying they could be
> > > replaced with other models, like neural networks. However, for probabilistic predictions on
> > > tabular data, we believe that the use of trees alone has substantial relevance, and that it is
> > > a meaningful contribution to the community for the following reasons:
> > >
> > > - *Trees are fast and easy to train.* Neural networks are slow to train and often require specialized
> > > hardware, especially within diffusion models. As mentioned in Section 1, Treeffuser trains from a table with 112,000
> > > observations and 27 variables in 53 seconds on a laptop.
> > >
> > > - *Trees make learning robust.* Neural networks are sensitive to the choice of architecture
> > > and training hyperparameters. Treeffuser outperforms existing methods across diverse
> > > datasets even without tuning.
> > >
> > > - *Trees are accurate on tabular data.* Trees have been shown to outperform neural
> > > networks on tabular data, including complex tabular data.
> > >
> > > We clarify that when we say that we adapt diffusions to gradient-boosted trees, we mean that we show concretely how to set up the learning objective (Theorem 2) and demonstrate empirically that this approach works. However, we do not claim to introduce a new diffusion framework. As discussed, we use the continuous-time formulation from Song et al. (2021), originally conceived for the unconditional case $p(x)$, and adapt it to our conditional setting $p(y|x)$ (Theorem 1).
> > >
> > >
> > > To conclude, we would like to emphasize once again that the use of trees is a central aspect
> > > of our contribution, and overlooking this might miss the essence of our work.

---

### Official Review · Reviewer_iWzb · 2024-07-12

**Soundness:** 3
**Presentation:** 4
**Contribution:** 3
**Rating:** 6
**Confidence:** 4

**Summary:**

This paper proposed a method called "Treeffuser" for probabilistic prediction (density regression) of tabular data.

**Strengths:**

The paper is well-written and clearly presented the core idea as well as the main results.

**Weaknesses:**

There should be more related methods to compare with in the experiment in section 5.3, especially more recent ones.

**Questions:**

What are the potential applications of the proposed method, especialy given that (1) it's designed for tabular data, and (2) it is generative but cannot provide density evaluation?

**Limitations:**

My main concern about this proposed method is that it cannot provide exact density evaluation. This is also discussed in the limitations section by the authors.

---

> ### Author Rebuttal · Authors · 2024-08-06
>
> # Rebuttal reviewer iWzb
> We thank reviewer iWzb for their comments. We appreciate that the reviewer found our paper to be well-written and clearly presented. We appreciate the great questions and suggestions by the reviewer. We provide answers and discussions below.
>
>
> ## Weaknesses
>
> **W1. There should be more related methods to compare with in the experiment in section 5.3, especially more recent ones.**
>
>
> We thank the reviewer for the suggestion. We designed Section 5.3 to be a concrete example showing how simple it is to use Treeffuser for real-world applications. This section was not intended to be an extensive comparison like Section 5.2.
>
> We selected the methods in Section 5.3 because they performed the best in Table 2, including the recent iBUG [1]. They also represent two types of methods with interesting properties for comparison with Treeffuser: a “hand-crafted” log-likelihood specially suited  for the problem (NGBoost Poisson), and a  “likelihood-free” tree-based model (amortized quantile regression). We omitted other methods in Figures 2 and 3 to enhance the figures’ legibility.
>
> ## Questions
> **Q1. What are the potential applications of the proposed method, especialy given that (1) it's designed for tabular data, and (2) it is generative but cannot provide density evaluation?**
>
> Thank you for your question:
> 1. Methods for predictions from tabular data are one of the most frequent uses of machine learning in practical industrial applications (e.g. sklearn [2], or methods such as xgboost [3]). Uncertainty estimation in the form of modeling $p(y \mid x)$ is equally important in those applications and for risk-aware decision making [4-7].
>
> 2. We do not focus on providing density evaluation because it is unnecessary for our purposes. In practice, we aim to model $p(y \mid x)$ to make real-world decisions based on quantities such as $\mathbb{E}[f(y, x) \mid x]$. These quantities are purely evaluated from samples of $y \mid x$. For example, a user might be interested in the expected output of their factory given specific settings, the standard deviation, the expected profit, or the probability that profit falls below a certain threshold $c$, where their profit function $f(y, x)$ is a black-box function. With Treeffuser, we can model and compute quantities by using the samples to compute $\mathbb{E}[y \mid x]$, $\mathbb{E}[y^2 \mid x] - \mathbb{E}[y \mid x]^2$, $\mathbb{E}[f(y, x) \mid x]$, and $\mathbb{E}[\mathbb{1}_{f(y, x) \leq c} \mid x]$, respectively. In contrast, with density estimation $p(y \mid x)$, evaluating $\mathbb{E}[f(y, x) \mid x]$ or even $\mathbb{E}[y \mid x]$ would likely require advanced techniques to sample from $p(y \mid x)$ and use Monte Carlo estimators to compute these quantities. Treeffuser performs these computations directly.
>
>
> To re-emphasize, while having an exact density evaluation could be useful in some scenarios, we believe having sampling is as important if not more important. In fact, the ability to sample from a density is an active research area (inverse transform sampling, rejection sampling), and Treeffuser enables sampling directly without modeling the density.
>
> We have now expanded the paper’s introduction with the discussion above to emphasize that the focus of the paper is sampling from $p(y \mid x)$ instead of approximating the value $p(y\mid x)$. Thank you for helping us clarify any confusion about the goal and impact of Treeffuser.
>
> ## Limitations
>
> **L1. My main concern about this proposed method is that it cannot provide exact density evaluation. This is also discussed in the limitations section by the authors.**
>
>
> As detailed above, we did not aim to do exact density evaluation because the downstream applications we are focusing on do not need it. We focused on designing a method that provides high-quality samples, which is more important in practice. We believe that focusing on sampling is a strength, not a weakness.

---

> > ### Comment · Reviewer_iWzb · 2024-08-13
> >
> > Thanks for the rebuttal. I keep my original score of weak accept.

---

### Author Rebuttal · Authors · 2024-08-06

# General rebuttal

We are grateful to the reviewers for their great feedback! Here, we summarize the key points from our rebuttals.

## Contributions

We clarify that our main contribution is to successfuly combine diffusion models and gradient-boosted trees (GBT), and use them to provide state-of-the-art probabilistic predictions on tabular data. Our approach combines the robustness and accuracy of GBT with the nonparametric flexibility of diffusion models.

Treeffuser is straightforward to use and widely applicable, as it comes with a plug-and-play Python package and requires little to no tuning.

Using Treeffuser with default parameters requires three lines of code:
```
model = Treeffuser()
model.fit(X, y)
samples = model.sample(n_samples)
```
Thanks to the use of diffusions and trees, this approach works with both multi-dimensional and uni-dimensional responses, regardless of whether $p(y \mid x)$ is multi-modal, heteroskedastic, heavy-tailed, or follows a simple normal distribution.

## Empirical experiments

We incorporated the reviewers feedback to expand our empirical experiments and strenghten our paper. More specifically,

 - We implemented a KDE-based version of IBUG and extended its hyperparameter search space. This improved the method’s performance, but Treeffuser still outperforms IBUG across most datasets.
 - We added vanilla XGBoost and LightGBM to the point prediction benchmark tables. As expected, they outperform or tie with all the probabilistic methods. In particular, they often tie with Treeffuser, suggesting that Treeffuser provides probabilistic prediction without sacrificing average point predictions.
 - We added an ablation study of Treeffuser where we removed the standard deviation reparametrization (line 133 and Eq. 13) and show that this design choice was important.
 - We conducted an empirical runtime analysis of the training and sampling processes. The results show that training is fast (less than 2 minutes for the datasets in the paper) and scales linearly with dataset size; sampling is also efficient ($\approx 10^{-4}$ seconds per sample) but slow for large sample sizes, yet it can be scaled with parallelization.


## Other feedback
We clarify that our focus is on generating samples rather than approximating the density $p(y|x)$. Practical applications require quantities such as expectations, standard deviations, quantiles, among others; while these quantities are hard to compute with densities, they can be readily computed with samples.

We also improved the clarity of our notation and derivations in our method section.

## References
These are the references cited in our rebuttals.

[1]  Jonathan Brophy, & Daniel Lowd. (2022). Instance-Based Uncertainty Estimation for Gradient-Boosted Regression Trees.

[2]  Pedregosa, F., Varoquaux, G., Gramfort, A., Michel, V., Thirion, B., Grisel, O., Blondel, M., Prettenhofer, P., Weiss, R., Dubourg, V., Vanderplas, J., Passos, A., Cournapeau, D., Brucher, M., Perrot, M., & Duchesnay, E. (2011). Scikit-learn: Machine Learning in Python. Journal of Machine Learning Research, 12, 2825–2830.

[3] Chen, T., & Guestrin, C. (2016). XGBoost: A Scalable Tree Boosting System. In Proceedings of the 22nd ACM SIGKDD International Conference on Knowledge Discovery and Data Mining. ACM.

[4] Gneiting, T., & Katzfuss, M. (2014). Probabilistic forecasting. Annual Review of Statistics and Its Application, 1(1), 125-151.

[5] Begoli, E., Bhattacharya, T., & Kusnezov, D. (2019). The need for uncertainty quantification in machine-assisted medical decision making. Nature Machine Intelligence, 1(1), 20-23.

[6] Jillian M. Clements, Di Xu, Nooshin Yousefi, & Dmitry Efimov. (2020). Sequential Deep Learning for Credit Risk Monitoring with Tabular Financial Data.

[7] Taylor, J. W., & Taylor, K. S. (2023). Combining probabilistic forecasts of COVID-19 mortality in the United States. European Journal of Operational Research, 304(1), 25–41.

[8] Yang Song, Jascha Sohl-Dickstein, Diederik P. Kingma, Abhishek Kumar, Stefano Ermon, & Ben Poole (2020). Score-Based Generative Modeling through Stochastic Differential Equations. CoRR, abs/2011.13456.

[9] Xizewen Han, Huangjie Zheng, & Mingyuan Zhou. (2022). CARD: Classification and Regression Diffusion Models.

[10] Borisov, V., Leemann, T., Seßler, K., Haug, J., Pawelczyk, M., & Kasneci, G. (2024). Deep Neural Networks and Tabular Data: A Survey. IEEE Transactions on Neural Networks and Learning Systems, 35(6), 7499–7519.

[11] Léo Grinsztajn, Edouard Oyallon, & Gaël Varoquaux. (2022). Why do tree-based models still outperform deep learning on tabular data?.

[12] Yury Gorishniy, Ivan Rubachev, Valentin Khrulkov, & Artem Babenko. (2023). Revisiting Deep Learning Models for Tabular Data.

[13] Jonathan Ho, Ajay Jain, & Pieter Abbeel. (2020). Denoising Diffusion Probabilistic Models.

[14] Jascha Sohl-Dickstein, Eric A. Weiss, Niru Maheswaranathan, & Surya Ganguli. (2015). Deep Unsupervised Learning using Nonequilibrium Thermodynamics.

[15] Ke, G., Meng, Q., Finley, T., Wang, T., Chen, W., Ma, W., Ye, Q., & Liu, T.Y. (2017). LightGBM: A Highly Efficient Gradient Boosting Decision Tree. In Advances in Neural Information Processing Systems. Curran Associates, Inc..

[16] Gneiting, T., & Raftery, A. E. (2007). Strictly proper scoring rules, prediction, and estimation. Journal of the American Statistical Association, 102(477), 359–378.

[17] Blicher Bjerregård, M., Kloppenborg Møller, J., & Madsen, H. (2021). An introduction to multivariate probabilistic forecast evaluation. Energy and AI, 4, 100058.

[18] Tony Duan, Anand Avati, Daisy Yi Ding, Khanh K. Thai, Sanjay Basu, Andrew Y. Ng, & Alejandro Schuler. (2020). NGBoost: Natural Gradient Boosting for Probabilistic Prediction.

---

### Decision · Program_Chairs · 2024-09-25

**Decision:**

Accept (poster)

**Comment:**

Treeffuser is a diffusion model in which the score function is approximated by gradient-boosted trees (unlike some other diffusion models that use neural networks).  Both the diffusion process and the GBTs are fairly standard, though the diffusion was adapted to the use of trees. The combination appears to be novel, and was clearly explained.  From the experimental results, it also seems to be quite effective for both synthetic and real-world tabular data sets, even if the hyper-parameters are not tuned.  There is also a software package which seems straightforward to use from the description.